# Last Glacial Maximum Climate and Atmospheric Circulation over the Australian Region from Climate Models

Yanxuan Du[1], Josephine R. Brown[1], J. M. Kale Sniderman[1]

[1] School of Geography, Earth and Atmospheric Sciences, University of Melbourne, Parkville, VIC, Australia

*Correspondence to*: Yanxuan Du (yanxuand@student.unimelb.edu.au)

**Abstract.** The Last Glacial Maximum (LGM, ~21,000 years ago) was the most recent time that the Earth experienced global maximum ice volume and minimum eustatic sea level. The climate changes over the Australian region at the LGM remain uncertain, including the extent of cooling in the arid interior, changes in the regional atmospheric circulations such as the tropical monsoon and mid-latitude westerlies as well as changes in the balance between precipitation and evaporation. In this

study, 13 climate model simulations that were included in the Paleoclimate Modelling Intercomparison Project (PMIP) Phases 3 and 4 are used to investigate regional climate (temperature, precipitation, and wind) over Australia at the LGM. The model simulations are compared with existing proxy records and other modelling studies. All models simulate consistent annual and seasonal cooling over the Australian region (defined as 0-45°S, 110°E-160°E) at the LGM compared to pre-industrial, with a multi-model mean 2.9 °C decrease in annual average surface air temperature over land at the LGM compared to pre-industrial.

Models simulate a range of LGM precipitation anomalies over the region. Simulated precipitation changes over tropical Australasia appear to be driven by changes in circulation and moisture transport, which vary greatly between models. Surface moisture balance calculated from precipitation minus evaporation shows little change over much of the Australian land area at the LGM. Changes in the strength and position of the mid-latitude westerlies are uncertain with wide model disagreement. These results indicate that climate model simulations do not show a robust response in either tropical or mid-latitude circulation

to LGM boundary conditions, suggesting that caution is required when interpreting model output in this region. Further analysis based on model evaluation and quantitative model-proxy comparison is required to better understand the drivers of LGM climate and atmospheric circulation changes in this region.

## 1 Introduction

The Last Glacial Maximum (LGM, ~21,000 years ago) refers to the coldest interval within the last glacial period, featuring

global maximum ice volume and associated low eustatic sea-level (Clark et al., 2009). Ice sheets covered large parts of North America, Europe, and Northern Eurasia at this time (Ehlers and Gibbard, 2007). In Australia, Reeves et al. (2013a) present evidence of small glaciers in the Snowy Mountains of southeast Australia and Tasmania, with the most significant glacier advance at around 19.1 ka (Petherick et al., 2013). Global mean surface air temperature (SAT) estimates during the LGM range from 3.4 to 8.3 °C cooler than pre-industrial based on different model ensemble results constrained with proxy data (Schneider

von Deimling et al., 2006; Holden et al., 2010; Annan and Hargreaves, 2013; Tierney et al., 2020a; Seltzer et al., 2021; Annan

et al., 2022) with recent studies suggesting a narrower cooling range of 4.5 °C ± 0.9 °C (Annan et al., 2022) or 6.1 °C ± 0.4 °C (Tierney et al., 2020a) depending on the method used. The global sea level was estimated at ca.120 meters lower than today (Lambeck et al., 2014; Yokoyama et al., 2018), resulting in the expansion of the land areas in many parts of the world, appearance of land bridges and the exposure of Sunda and Sahul shelves in Southeast Asia, which further allowed human migration during this period (Clarkson et al., 2017). The Australian mainland expanded at the LGM, connected by land bridges to New Guinea, Tasmania, and many smaller islands into a single landmass known as "Sahul" (Clarkson et al., 2017).

LGM vegetation was also very different from today. There was a large reduction in area covered by boreal and temperate forests in northern mid- to high latitudes, expanded lowland tundra in Eurasia, expansion of savanna and grasslands at the margin of Amazon tropical forests and replacement of some areas of tropical forest in Africa, China and Southeast Asia with savanna, woodland and grassland (Prentice et al., 2011). These changes are associated with large LGM reduction in terrestrial biomass, caused by a combination of lower temperatures, changes in hydroclimate (moisture availability), and/or lower atmospheric $CO_2$ concentrations (~180 ppm) that restricted vegetation growth (Scheff et al., 2017; Prentice et al., 2022). Dust suspension and transport was estimated to be more active at the LGM, which was possibly driven by stronger winds and drier climate conditions, particularly in the tropical and high-latitude regions (Lamy et al., 2014), or may have been partly a consequence of the reduced ability of terrestrial vegetation to stabilize soils (Scheff et al., 2017; Roderick et al., 2015).

While many modelling studies have focused on the LGM climate globally and in the Northern Hemisphere (e.g. Tierney et al., 2021a; Kageyama et al., 2021), LGM changes in Southern Hemisphere (SH) regional temperature, rainfall and atmospheric circulation are less well understood, including for Australia. Questions about the climate of Australia at the LGM include the extent of cooling in the arid interior, changes in the regional atmospheric circulations such as the tropical monsoon and mid-latitude westerlies, as well as changes in the balance between precipitation and evaporation. Understanding the changes in regional climate over Australia at the LGM is of interest for a number of reasons. The Australian climate is influenced by major tropical and extratropical modes of circulation, including the Walker circulation and SH Hadley Cell, Indo-Australian monsoon and mid-latitude westerlies. Therefore, understanding how the climate of Australia changed at the LGM could provide insight into the changes in these large-scale climate processes, which may also be relevant for constraining uncertainty in future changes in the region (e.g. Grose et al., 2020; Narsey et al., 2020). In addition, it is also crucial for interpreting records of human occupation (Williams et al., 2013; Bird et al., 2016; Bradshaw et al., 2021) and changes in flora and fauna distributions (Byrne et al., 2008; Nevill et al., 2010; Pepper and Keogh, 2021) from this period.

A number of studies have provided comprehensive synthesis of available terrestrial and marine records over the SH during the LGM (e.g. Reeves et al., 2013a; Reeves et al., 2013b; Petherick et al., 2022). However, limited numbers of climate modelling studies have focused on LGM conditions in the SH (e.g. Rojas et al., 2009; Rojas, 2013; Sime et al., 2013), with even fewer studies examining simulations of LGM climate in Australia (e.g. Hope, 2005; Yan et al., 2018). Some studies (e.g. Brown et al., 2020; Kageyama et al., 2021; Gray et al., 2023; Wang et al., 2023) have explored the new PMIP4 simulations of LGM climate, but there has been little research on the changes in SH climate, including the Australian region in these simulations (or the older PMIP3 ensemble). This study investigates the Australian regional climate changes at the LGM using

available simulations from PMIP3/PMIP4 models and compares the results with existing proxy records and other model studies

of the region. The aim is to provide preliminary insights into the LGM climate in Australia from model simulations and evaluate

the model consistency with available proxy records.

## 1.1 Palaeoenvironmental proxy records for the Australian LGM

This section summarises proxy records across the Australian region separated into three distinct climate zones: tropical

northern Australia, temperature southern Australia and arid central Australia, consistent with previous classifications (e.g.

Reeves et al., 2013a; Petherick et al., 2013; Fitzsimmons et al., 2013). The regions covered by these climate zones are shown

in Figure 1, with indications of the locations of proxy records discussed below where available. Proxy records are included

where dating falls within the period 24 to 18 ka, considered broadly representative of LGM reconstructed climate in Australia

(see Cadd et al., 2021 and Petherick et al., 2022 for further discussion of the timing of the LGM in Australia). These proxy

records are compared with our model simulations in Section 4.

In northern Australia (see Figure 1 for corresponding region), surrounding sea surface temperatures (SSTs) are

estimated to have cooled by 1 to 3 °C at the LGM relative to present (Reeves et al., 2013a). The temperature in upland areas

in New Guinea is estimated to have reduced by 4 to 6 °C compared to present based on pollen records from near treeline in

the Kosipe Valley (Hope, 2009). A dramatic reduction in tree cover was identified at the LGM in tropical savanna woodland

in northern Australia from pollen and geochemical records (Rowe et al., 2021). Rowe et al. (2021) attributed the change in

vegetation to a combination of a cooler and drier glacial climate, while also considering the possible role of lower atmospheric

$CO_2$ concentrations and the increased distance of the site from the coastline. Reduced fire activity occurred as a result of less

available fuel, indicating that the vegetation at the LGM was less influenced by fire events than today (Rowe et al., 2021).

There is uncertainty about the drivers of LGM climate changes in northern Australia; sparser vegetation has been interpreted

previously as a sign of aridity, but it is also likely that low $CO_2$ played a role in reducing plant biomass (e.g. Scheff et al., 2017;

Prentice et al., 2017; Prentice et al., 2022). Denniston et al. (2013) found evidence of an active but variable monsoon during

the LGM based on speleothem records at Ball Gown Cave in tropical northern Australia, with more positive speleothem

isotopic values at the LGM than the late Holocene indicating relatively dry glacial conditions.

In southern Australia during the LGM, a SAT reduction of 4-6 °C has been inferred from pollen records, and SST

cooling varying from 3 to 9 °C has been inferred in nearby ocean regions (Petherick et al., 2013). Fossil pollen records

indicating widespread reductions in tree cover have often been interpreted as implying drier conditions, i.e. reduced

precipitation and/or increased evaporation (Petherick et al., 2013), but the potential role of low atmospheric $CO_2$ in reducing

plant productivity has rarely been considered in the region (Prentice et al., 2017; Sniderman et al., 2019). Moreover, regionally

wetter conditions (increased precipitation and/or reduced evaporation) were also present in some parts of the southern

Australian domain (Reeves et al., 2013b), perhaps consistent with an equatorward shift of the SH westerlies at the LGM

(Kohfeld et al., 2013). Some evidence of higher lake (Lakes Mungo, Keilambete and George) and river levels in the Murray-

Darling Basin (Hesse et al., 2018) has been interpreted in terms of greater seasonal runoff due to snowmelt (Petherick et al., 2013), or some combination of higher precipitation and lower evapotranspiration (Hesse et al., 2018). In subtropical eastern Australia, the persistence of moisture-demanding woodlands suggests that the effective precipitation (net moisture) levels did
not drop dramatically during the LGM in this region (Cadd et al., 2018).

In central Australia, the average LGM air temperature in the arid zone was estimated to decrease by 9 °C below present (Miller et al., 1997) based on amino-acid racemisation of emu eggshell. Fitzsimmons et al. (2013) argued that the arid interior experienced extensive dune activity and dust transport, reduced but episodic fluvial activity at the LGM relative to pre-industrial (PI). However, there is geomorphological evidence for higher lake levels at Lake Frome (Cohen et al., 2015).
Evidence of wetter conditions in arid Australia during the LGM was found by Treble et al. (2017) from speleothem records at Mairs Cave, Flinders Ranges in the southern Australian semi-arid zone. That study suggested that the Flinders Ranges were relatively wet during the LGM, possibly associated with a southward shift of the Intertropical Convergence Zone (ITCZ), allowing more tropical moisture to reach the cave. These proxy records providing evidence of relatively wet conditions at the LGM support the hypothesis proposed by De Deckker et al. (2020), that water was available during the cold and dry LGM
period in Australia, sustaining human populations in inland areas.

In summary, there is evidence for widespread cooling over northern, southern and central Australia at the LGM based on a range of proxy records, with the largest cooling inland and at high elevations. In northern Australia, vegetation and fire records suggest drier conditions or possible influences from lower $CO_2$, while speleothem records may indicate drier conditions. Records from southern and central Australia provide uncertain evidence for hydroclimate change, with some
records supporting wetter conditions and others implying drying. Moisture changes may reflect meridional shifts in the westerlies or tropical convergence zone, but evidence for these shifts is also contradictory.

**1.2 Climate models simulations of the LGM**

In this context, climate models could thus provide insights into the mechanisms responsible for this observed climate. Many previous modelling studies have focused on the LGM (Kageyama et al., 2017), however few studies have examined the SH or
the Australian region, as noted above. In this section, we summarise previous LGM modelling studies which focus on the Australian region and major modes of circulation including the Indo-Australian monsoon and the SH mid-latitude westerlies.

Several early studies made use of low-resolution climate models to examine the changes in the Australian monsoon over the late Quaternary (e.g. Wyrwoll and Valdes, 2003; Marshall and Lynch, 2006). In particular, Marshall and Lynch (2006) employed the FOAM model to simulate time-slices including the LGM and found that there was an overall drying over northern
Australia at the LGM, with slightly increased precipitation during monsoon onset offset by drying during the monsoon peak and retreat. More recently, Yan et al. (2018) examined the Australian monsoon in PMIP3 simulations and found decreased annual and winter precipitation over northern Australia with increases in the early summer (November to December) due to enhanced moisture convergence. Changes in northern Australian temperature, precipitation and atmospheric circulation are

further examined in this study with a larger ensemble of models including five models from the latest CMIP6-PMIP4 generation.

The behaviour of the SH westerlies at the LGM is a major area of research with relevance for southern Australian climate. Previous modelling studies using coupled atmosphere-ocean models (PMIP2 and 3) and atmosphere-only models (PMIP1) show ambiguous results regarding the latitudinal positions of the SH mid-latitude westerlies during the LGM. For example, Sime et al. (2013) found strengthening and poleward shifts in the maximum 850 hPa SH westerlies based on PMIP2 simulations with an atmosphere-only model (HadAM3). Similarly, a poleward shift in SH surface westerlies was indicated by Kitoh et al. (2001) from an AOGCM used in PMIP1 (MRI-CGCM1 model). In contrast, Gray et al. (2023) proposed a new method with PMIP3/4 model ensemble and compiled planktic foraminiferal $\delta^{18}O$ data from the Southern Ocean, suggesting equatorward and weakened SH westerlies at the LGM. Disagreements are found between different models, with equatorward shifts (Kim et al., 2003) and no latitudinal change (Otto-Bliesner et al., 2006; Rojas et al., 2009) in SH westerlies also observed from PMIP model simulations. This disagreement across model simulations is consistent with Chavaillaz et al. (2013) using the PMIP3 and CMIP5 models, suggesting no agreement was reached between models regarding the latitudinal changes in SH westerlies during the LGM. Rojas (2013) suggested that differences in response may be related to the coupling between atmosphere, ocean and sea ice in the models. To further test model agreements, shifts in SH westerlies and their influences on southern and central Australian climate in an ensemble of CMIP5-PMIP3 and CMIP6-PMIP4 models are investigated in this study.

Utilising LGM simulations from an ensemble of CMIP5-PMIP3 and CMIP6-PMIP4 models, this study provides a summary of changes in temperature, precipitation and winds over the Australian region, including both multi-model mean changes and evaluation of model spread or agreement. The drivers of precipitation changes are explored, with reference to changes in regional temperature and circulation patterns. Simulated changes in the summer monsoon and the mid-latitude westerlies are also examined. The model simulations are compared with proxy reconstructions of temperature and precipitation or moisture balance to determine the extent of model-proxy agreement.

In this paper, the data and methods are described in Section 2, Section 3.1 presents temperature results from models, while wind and precipitation changes at the LGM relative to PI are shown in Section 3.2 and 3.3, respectively. Relationships between climate variables, such as SH mid-latitude westerly winds and precipitation changes at the LGM in austral winter (JJA) season, and the correlations between seasonal temperature and precipitation patterns are evaluated in Section 3.3.2 as drivers of precipitation change. Section 4 discusses the limitations and consistencies between relevant proxy records and modelling studies with our model results, followed by the conclusion in Section 4.4.

## 2 Data and methods

### 2.1 Model datasets

This study makes use of PMIP Phase 3 (PMIP3, Braconnot et al., 2012) model simulations which were included in the Coupled Model Intercomparison Project Phase 5 (CMIP5, Taylor et al., 2012), and PMIP Phase 4 (PMIP4, Kageyama et al., 2018) simulations which were included in CMIP Phase 6 (CMIP6, Eyring et al., 2016). Datasets from eight CMIP5 models and five CMIP6 models that were included in PMIP3 and PMIP4 were analysed, based on data availability via the Earth System Grid Federation (ESGF, the set of models is therefore smaller than Kageyama et al. (2021), who made use of some models only

available in PMIP databases). See Table 1 for list of models with LGM simulations included in this study. Monthly surface temperature (ts), surface air temperature (tas), precipitation (pr), evapotranspiration (evspsbl) and 850 hPa wind (ua and va at 850 hPa) data from each model for a model pre-industrial control ('piControl') and LGM ('lgm') simulations were analysed. These variables were chosen to characterise the main features of LGM climate in the region and to facilitate comparison with proxy records of LGM temperature and hydroclimate. All data was re-gridded using first-order conservative remapping onto

a $1.5° \times 1.5°$ longitude-latitude grid. Both zonal (u) and meridional (v) components of the wind were analysed, at 850 hPa except for CMIP6 INM-CM4-8 model, which only provided near-surface (10 m) wind data for LGM simulations. PMIP3/CMIP5 and PMIP4/CMIP6 models shown in Table 1 are referred to as CMIP5 and CMIP6 models hereafter for simplification.

**Table 1:** List of models with LGM data on NCI analysed in this study. The ice-sheet reconstructions are from Kageyama et al. (2021).

| Model | PMIP/CMIP gen. | Ice-sheet reconstruction | Reference |
|---|---|---|---|
| AWI-ESM-1-1-LR | PMIP4 (CMIP6) | ICE-6G_C | Sidorenko et al. (2015), Lohmann et al. (2020) |
| CCSM4 | PMIP3 (CMIP5) | PMIP3 | Brady et al. (2013) |
| CESM2-WACCM-FV2 | PMIP4 (CMIP6) | ICE-6G_C | Zhu et al. (2021) |
| CNRM-CM5 | PMIP3 (CMIP5) | PMIP3 | Voldoire et al. (2013) |
| FGOALS-g2 | PMIP3 (CMIP5) | PMIP3 | Zheng and Yu (2013) |
| GISS-E2-R | PMIP3 (CMIP5) | PMIP3 | Ullman et al. (2014) |
| INM-CM4-8 | PMIP4 (CMIP6) | ICE-6G_C | Volodin et al. (2018) |

| IPSL-CM5A-LR | PMIP3 (CMIP5) | PMIP3 | Dufresne et al. (2013) |
|---|---|---|---|
| MIROC-ES2L | PMIP4 (CMIP6) | ICE-6G_C | Ohgaito et al. (2021), Hajima et al. (2020) |
| MIROC-ESM | PMIP3 (CMIP5) | PMIP3 | Sueyoshi et al. (2013) |
| MPI-ESM-P | PMIP3 (CMIP5) | PMIP3 | Adloff et al. (2018) |
| MPI-ESM1-2-LR | PMIP4 (CMIP6) | ICE-6G_C | Mauritsen et al. (2019) |
| MRI-CGCM3 | PMIP3 (CMIP5) | PMIP3 | Yukimoto et al. (2015) |

The models in CMIP6 include some modifications and improvements relative to the older CMIP5 generation (Eyring et al., 2016). There are also some differences in the LGM experiment boundary conditions for the CMIP5 (PMIP3) and CMIP6 (PMIP4) experiments. Compared to PMIP3 experiments, new and updated boundary conditions were included in PMIP4 (as shown in Table 2), enabling the systematic analysis of the vegetation and dust forcing effects. Furthermore, the PMIP4 protocol highlighted the specification of ice sheets, with three distinct ice sheet reconstructions available, allowing assessments of the impacts from uncertainties in ice-sheet reconstructions or boundary conditions (Kageyama et al., 2017). The PMIP3 models used the PMIP3 ice-sheet configuration and the five PMIP4 models used in this study are all prescribed using the "ICE-6G_C" ice-sheet reconstruction (see Table 1). The differences between the two LGM ice-sheet reconstructions are discussed in Kageyama et al. (2017). Only CMIP6 AWI-ESM-1-1-LR model is using dynamic vegetation (https://wcrp-cmip.github.io/CMIP6_CVs/docs/CMIP6_source_id.html). The different LGM coastlines configured in each model can be seen in Supplementary Figure S1 showing the land masks for individual models. The configuration of the expanded Sahul shelf varies widely between models.

**Table 2:** Summary of the main forcing or boundary conditions (experimental design) for the LGM simulations in PMIP3 and PMIP4 models (PMIP4 from Table 1 in Kageyama et al. (2017); PMIP3 from PMIP3 website: https://pmip3.lsce.ipsl.fr/). Some boundary conditions are set as the same as pre-industrial control (piControl) values.

| Forcing or Boundary conditions | PMIP4 LGM value | PMIP3 LGM value |
|---|---|---|
| Atmospheric trace gases | $CO_2$ = 190 ppm<br>$CH_4$ = 375 ppb<br>$N_2O$ = 200 ppb | $CO_2$ = 185 ppm<br>$CH_4$ = 350 ppb |

|  | CFC = 0<br>O$_3$ = same as in CMIP6 *piControl* | N$_2$O = 200 ppb<br>CFC = 0<br>O$_3$ = same as in CMIP5 *piControl* |
|---|---|---|
| Insolation | eccentricity: 0.018994<br>obliquity: 22.949°<br>perihelion − 180° = 114.42° | eccentricity: 0.018994<br>obliquity: 22.949°<br>perihelion − 180° = 114.42° |
| Ice sheets (components of model modified to represent influence of LGM ice sheet) | coastlines<br>bathymetry ice-sheet extent<br>altitude<br>rivers | land-sea mask<br>land surface elevation<br>ocean bathymetry |
| Vegetation | Unless a model includes dynamic vegetation or interactive dust, the vegetation should be prescribed to be the same as in the DECK and historical runs (CMIP6 *piControl*) | as in *piControl* |
| Dust | as in *piControl*<br>or<br>*lgm* (three options) | as in *piControl* |

## 2.2 LGM and Control Simulations

The LGM simulations are compared with pre-industrial (PI) climate as the control or baseline, similarly to many previous studies (e.g. Kageyama et al., 2021). Pre-industrial ("piControl") experiments are simulations with atmospheric composition and other boundary conditions prescribed and held constant at values representing climate before industrialization, i.e. reference year 1850 (Eyring et al., 2016). Most climate models used in this study only have 100-year length of LGM simulation based on the number of years of data available on the ESGF. According to the PMIP protocols, the models need to spin-up to equilibrium before uploading the data to ESGF (see Kageyama et al. (2017) for details of the spin-up protocol and Kageyama et al. (2021) for PMIP4 model spin-up durations). At least 100-year data from the equilibrium part of the simulation is required to store on the ESGF (Kageyama et al., 2017). In this research, the first 100 years of output on the ESGF for each model was selected and averaged in order to capture the average climatology for each model simulation and to represent the mean state of PI and LGM climate conditions.

## 2.3 Classification of Australian Regions

Different regions of Australia experience different climate regimes in the present day, and are likely to respond differently to LGM climate forcing. Each region also experiences greater seasonal precipitation at a different time of year, with the north receiving most precipitation in austral summer (DJF season) and the south in austral winter (JJA season). Austral summer is the period when the Indo-Australian summer monsoon is most active, bringing precipitation to Northern Australia and nearby SH Maritime Continent areas. In austral winter, the SH westerlies shift equatorward, bringing rainfall to the Southern Australian region. We therefore divide Australia into three main regions in order to examine the LGM climate response in detail at seasonal time scales.

Shown in Figure 1, the Australian region is defined in this research bounded by longitudes from 110°E to 160°E, latitudes from 0 to 45°S. The Northern Australasia domain in the spatial plots is defined longitudes from 110°E to 160° E, and latitudes from 0 to 20°S (including New Guinea and parts of Indonesia); the Southern Australia domain is defined by longitudes from 110°E to 160°E, and latitudes from 20°S to 45°S. Lastly, the Central Australian domain is shown in the spatial plots bounded by 20°S to 35°S and 120°E to 145°E.

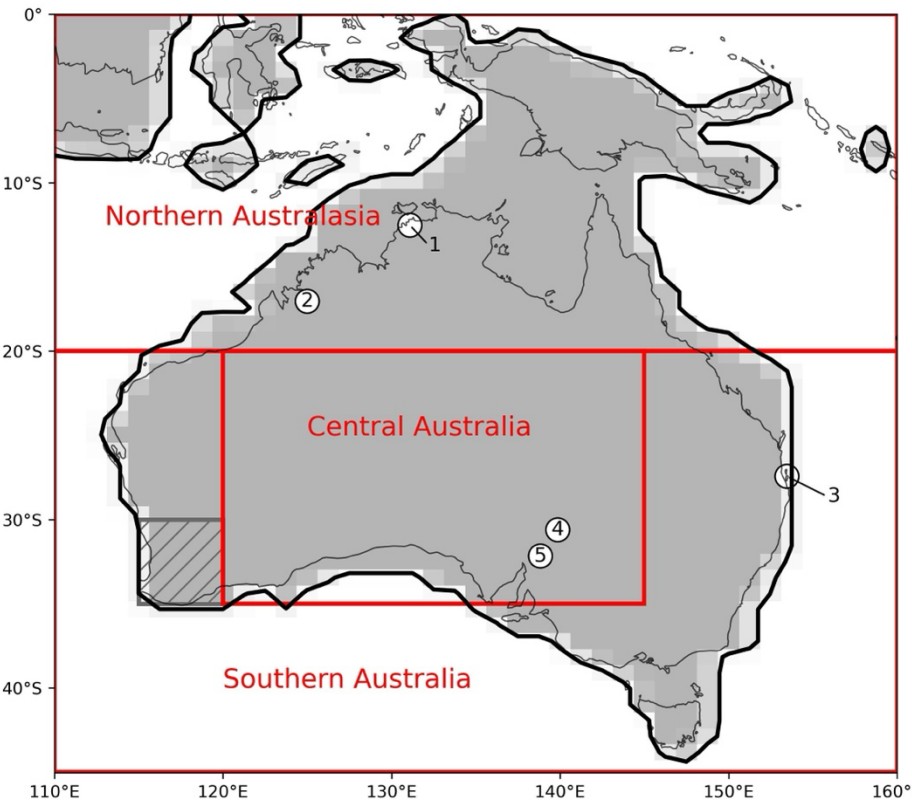

**Figure 1:** Classification of Northern Australasia, Central, and Southern Australia domains shown by red rectangles with modern day coastlines. The Central Australia domain is a subset of the larger Southern Australia domain. The south-west box for detecting JJA frontal

 precipitation that is most sensitive to westerlies is also indicated in hashes in the Southern Australia domain. The background grey shading indicates LGM land mask from CCSM4 model, the thick black contour denotes the LGM coastline, plotted over modern coastlines in thin black lines. Proxy records with specific locations discussed in the study are displayed in dots with numbering: (1) Rowe et al., 2021; (2) Denniston et al., 2013; (3) Cadd et al., 2018; (4) Cohen et al., 2015; (5) Treble et al., 2017, see Section 1.1 for detailed discussion.

## 3 Results

### 3.1 Surface air temperature

In this section, the changes in surface air temperature (tas) in the LGM model simulations are evaluated in comparison with PI simulations. Figure 2 shows the simulated surface air temperature patterns from CMIP5 and CMIP6 model ensembles. While stippling is used to indicate regions where less than 70% of ensemble members agree on the sign of the anomaly in multi-model mean (MMM) figures, no stippling is shown in Figure 2 due to the high model agreement. Despite some differences between the CMIP6 and CMIP5 model simulations in LGM temperature over Australia (Figure 2c), we combine the five models from CMIP6 and the eight models from CMIP5 together into a large ensemble of thirteen models, for assessing the MMM change in temperature patterns at the LGM in this section. Moreover, the MMM PMIP3 (CMIP5) and PMIP4 (CMIP6) Australian annual, seasonal and global averages are relatively similar to each other, despite some outliers (Figure 3).

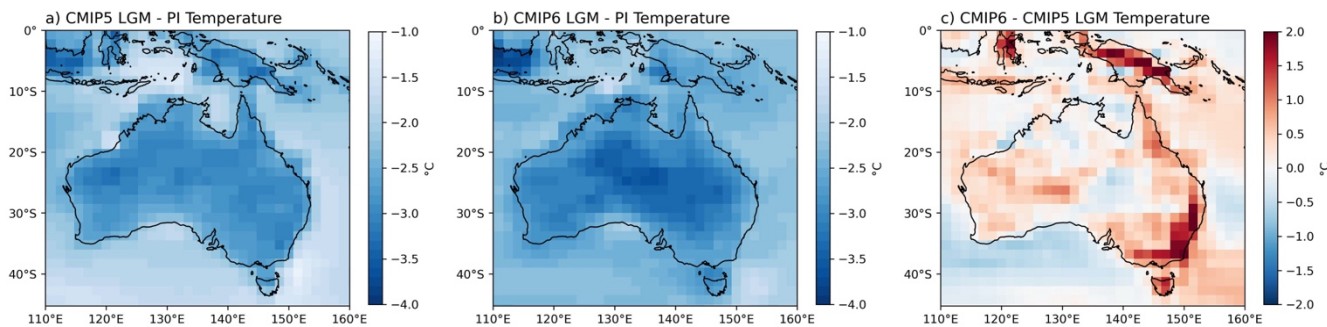

**Figure 2:** (a) LGM – PI mean annual surface air temperature (in °C) simulated by the ensemble of CMIP5 models, (b) LGM – PI mean annual surface temperature anomaly (in °C) simulated by CMIP6 models, (c) difference between the CMIP6 and CMIP5 LGM ensembles (in °C) over Australian region. Stippling indicates areas where less than 70% of ensemble members agree on the sign of the anomaly.

The area average changes in surface air temperature over Australia are shown for annual, DJF and JJA seasons, and compared with global annual average changes for each model (Figure 3). It is evident that there is slightly more cooling over Australia in JJA than DJF in the PMIP3 and PMIP4 MMM, indicating an enhanced seasonal cycle (although not all models show this). The cooling in CESM2-WACCM-FV2 model is much greater than the other models in all seasons and at both the Australian and global scales, while CNRM-CM5 has much less cooling than other PMIP3 (CMIP5) models at the global scale but only slightly less over Australia. Overall, the global temperature changes do not clearly correlate with regional changes over Australia from the same model.

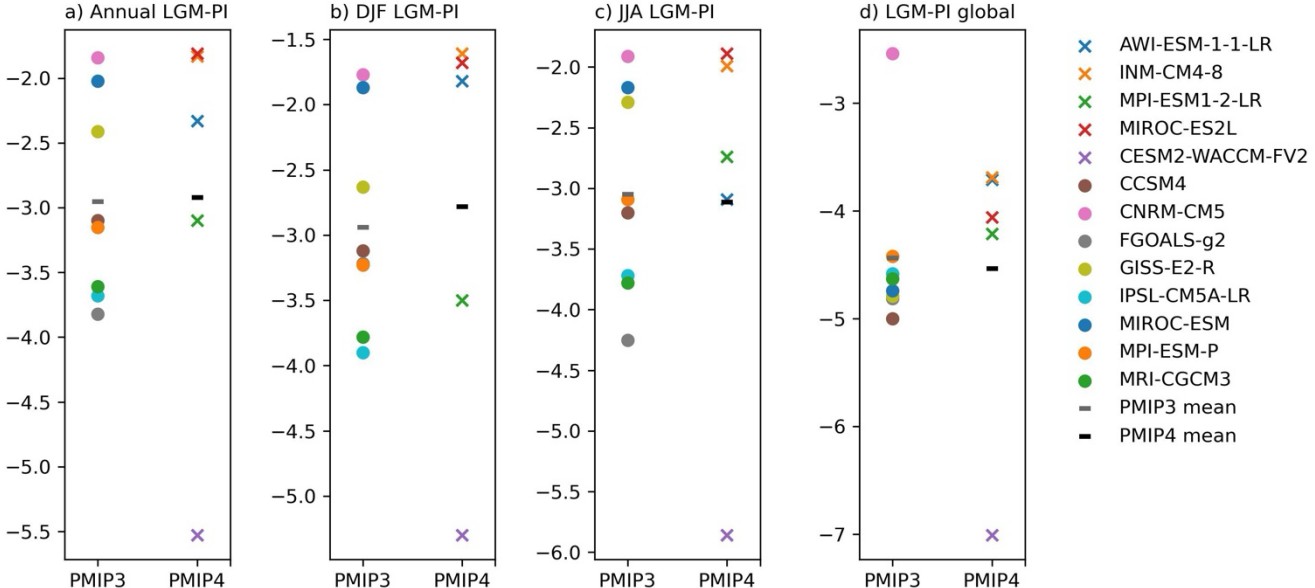

**Figure 3:** LGM – PI surface air temperature (tas) anomalies (°C) for (a) annual mean, (b) DJF, (c) JJA seasons over Australia (land areas within domain 0-45°S, 110°E-160°E), and (d) global annual mean surface air temperature anomalies (°C) for each PMIP3 and PMIP4 model and the multi-model mean (MMM).

To explore the extent of model agreement for LGM surface air temperatures, Figure 4 shows the annual mean LGM - PI surface air temperature changes over Australia in individual models, with the thick black lines showing the prescribed LGM coastlines in each model and the thin black lines indicate modern coastlines. All models agree on the sign of temperature change over the Australian region, with mean annual cooling during the LGM compared to PI. The cooling patterns are overall similar across most models, with more cooling over land than ocean. The MMM annual cooling averaged over Australian land areas (using modern coastlines) is -2.9 °C. Those models with greater cooling over the Australian region also tend to have greater global cooling (see Figure 3a and d). We note the much larger cooling in CMIP6 CESM2-WACCM-FV2 model is due to higher climate sensitivity of this model (Zhu et al., 2022), as seen in Figure 3d.

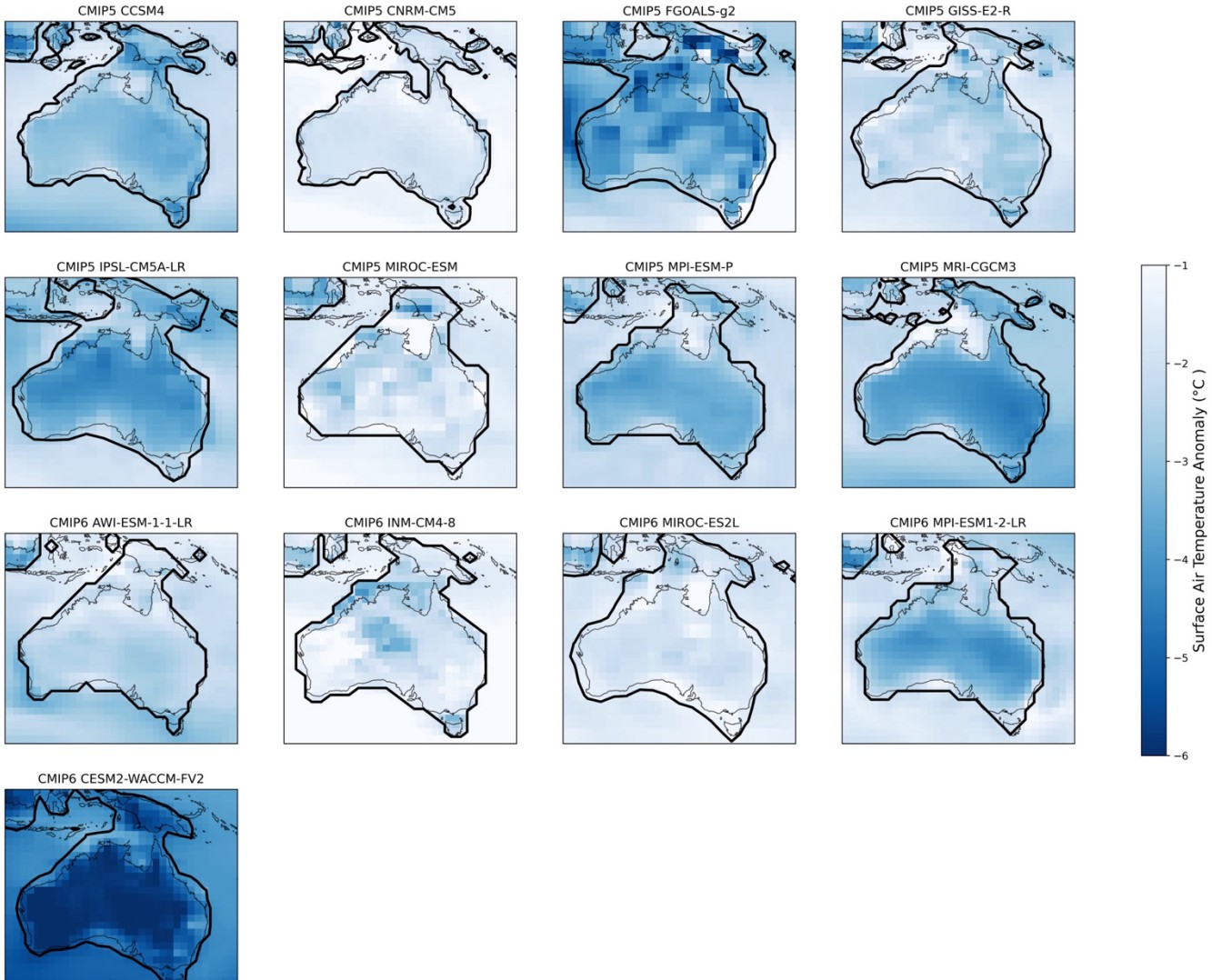

**Figure 4:** LGM - PI mean annual surface air temperature anomaly (°C) simulated by individual CMIP5 and CMIP6 models over Australian region. The thick black lines denote the coastlines in LGM prescribed in each model, and the thin black lines denoted the coastlines in piControl. 30% land area fraction was selected for plotting the LGM land mask contours in models.

The MMM seasonal variations in surface air temperature changes at the LGM over Australia are shown in Figure 5. The austral winter JJA season shows the strongest land average cooling (-3.1 °C, refer to Figure 3c) at the LGM, particularly along the modern coastlines in both the Northern and Southern Australian regions. LGM cooling is generally larger over land than the surrounding ocean in all seasons, with the exposed Sahul shelf area showing enhanced cooling in all seasons except SON, when there is model disagreement over the sign of the temperature anomaly over Sahul (see Figure 5d).

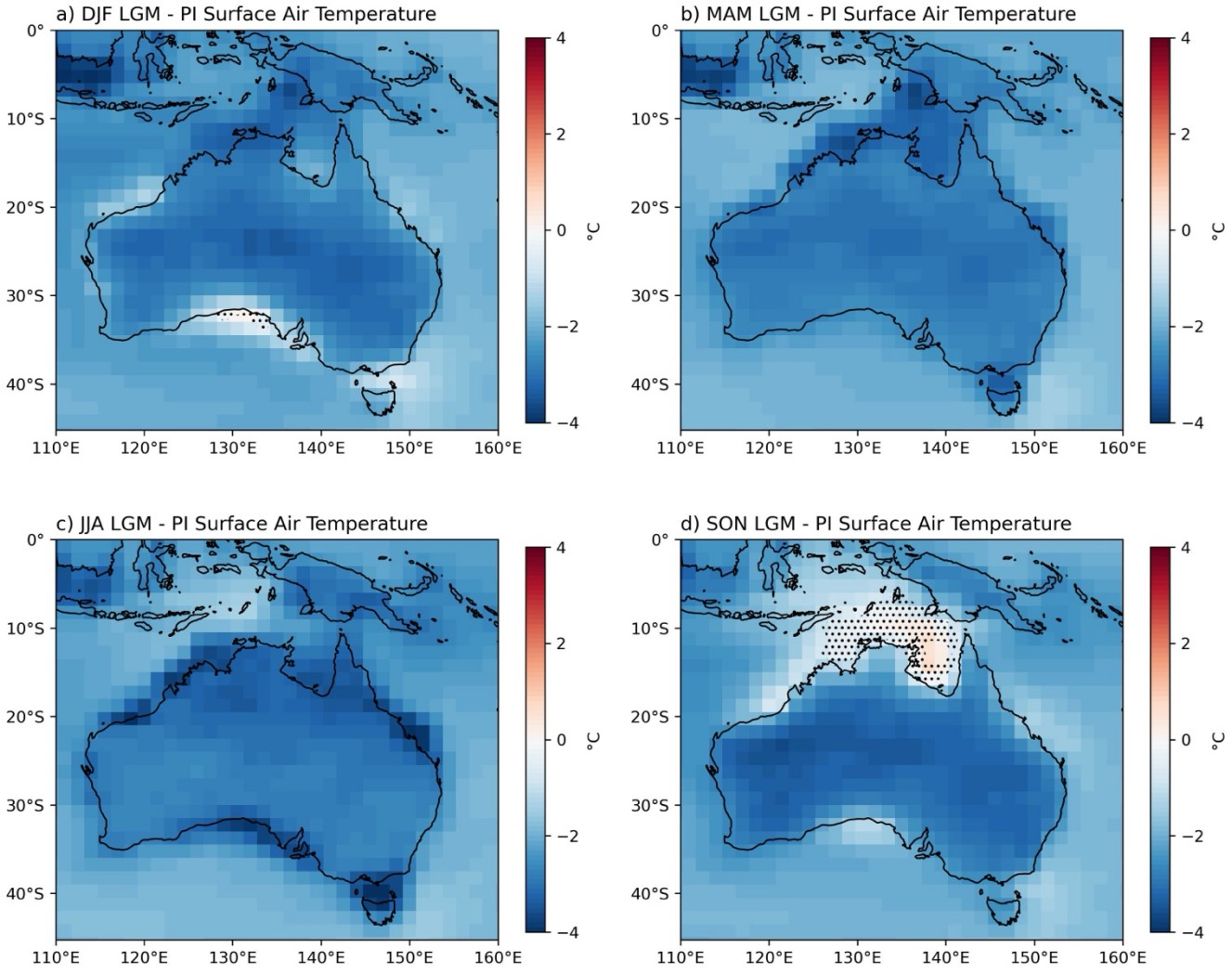

**Figure 5:** MMM seasonal anomalies for LGM - PI surface air temperatures (°C) simulated by the ensemble of CMIP5 and CMIP6 models for (a) DJF, (b) MAM, (c) JJA and (d) SON seasons over Australian region. Stippling indicates areas where less than 70% of ensemble members agree on the sign of the anomaly.

## 3.2 Winds

The previous section showed relatively consistent cooling patterns for Australian surface air temperature changes at the LGM simulated by models. This section explores the LGM - PI 850 hPa wind patterns and associated circulation changes from CMIP5/CMIP6 models over the Australian region. MMM seasonal anomalies for LGM - PI 850 hPa zonal winds and wind divergence over Australia are shown in Figure 6 (MMM excludes CMIP6 INM-CM4-8 model, with only 10 m near-surface

winds available). We first consider changes over the tropical Northern Australasia domain and then focus on the temperate

Southern Australia domain.

In austral summer (DJF) in the modern climate, the Indo-Australian summer monsoon is most active, with north-westerly winds and onshore moisture transport. In the LGM simulations there is a strengthening of the westerly component of the monsoon flow over Northern Australia during DJF (Figure 6a). There are regions of increased wind divergence (orange colors in Figure 6) over the Sahul shelf and parts of the expanded LGM coastline of northwest Australia, with regions of

convergence or weakened divergence over adjacent ocean areas. A similar pattern of enhanced onshore westerly flow and wind convergence upstream of the Sahul shelf is seen in austral spring (SON, Figure 6d). Strengthened divergence and offshore flow is seen over this region in MAM (Figure 6c).

In austral winter (JJA) in the modern climate, the SH mid-latitude westerly winds shift equatorward, allowing extratropical weather systems to influence Southern Australia. In the LGM simulations, the MMM change in JJA winds (Figure

6c) shows a weakening of the westerlies to the south of around 35°S with little change over Southern Australian land areas, with the exception of a small increase in the westerly winds and wind convergence over the south-west part of this domain. Individual model simulation results are discussed next to assess the extent of model agreement in JJA season wind changes.

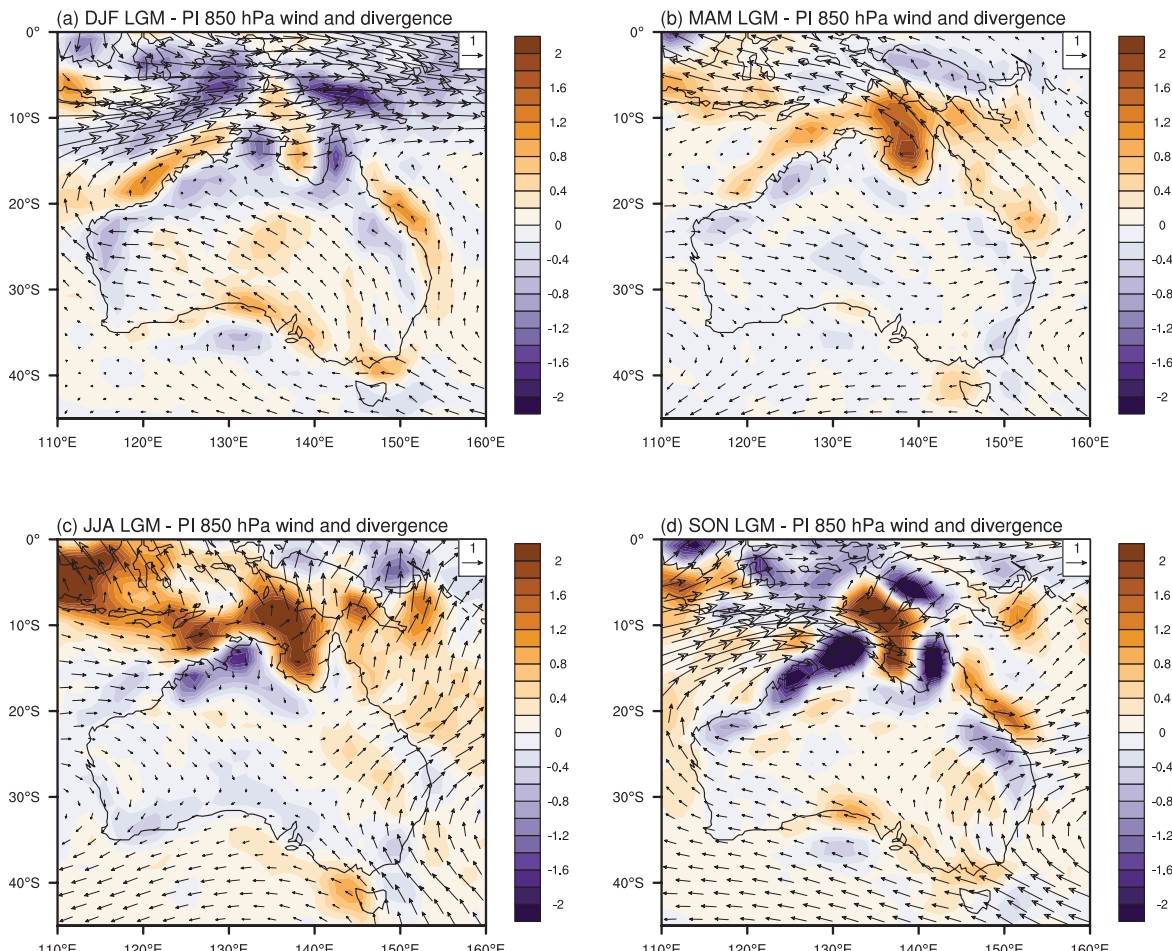

**Figure 6:** Seasonal average multi-model mean change (lgm – piControl) in 850 hPa winds (m/s) and wind divergence (1 x 10⁻⁶/s). Wind change is shown as vectors and wind divergence change is shown as colored contours. MMM excludes INM-CM4-8 model which provided only near-surface winds.

We next investigate the meridional displacement of the SH mid-latitude westerly winds at the LGM, in comparison with previous studies (see Section 1.2), as these winds play an important role in the climate of Southern Australia (e.g. Hope et al., 2010). The LGM change in mean annual winds for individual models is shown in Figure 7, with the change in the zonal component of the wind indicated as color shading. It is evident that there is large model disagreement over changes in the mid-latitude westerlies in the Australian region. Some models show weakening and other models show strengthening in the region to the south of Australia, with a slight weakening of the MMM westerly wind south of around 35°S as discussed above (see Figure 6).

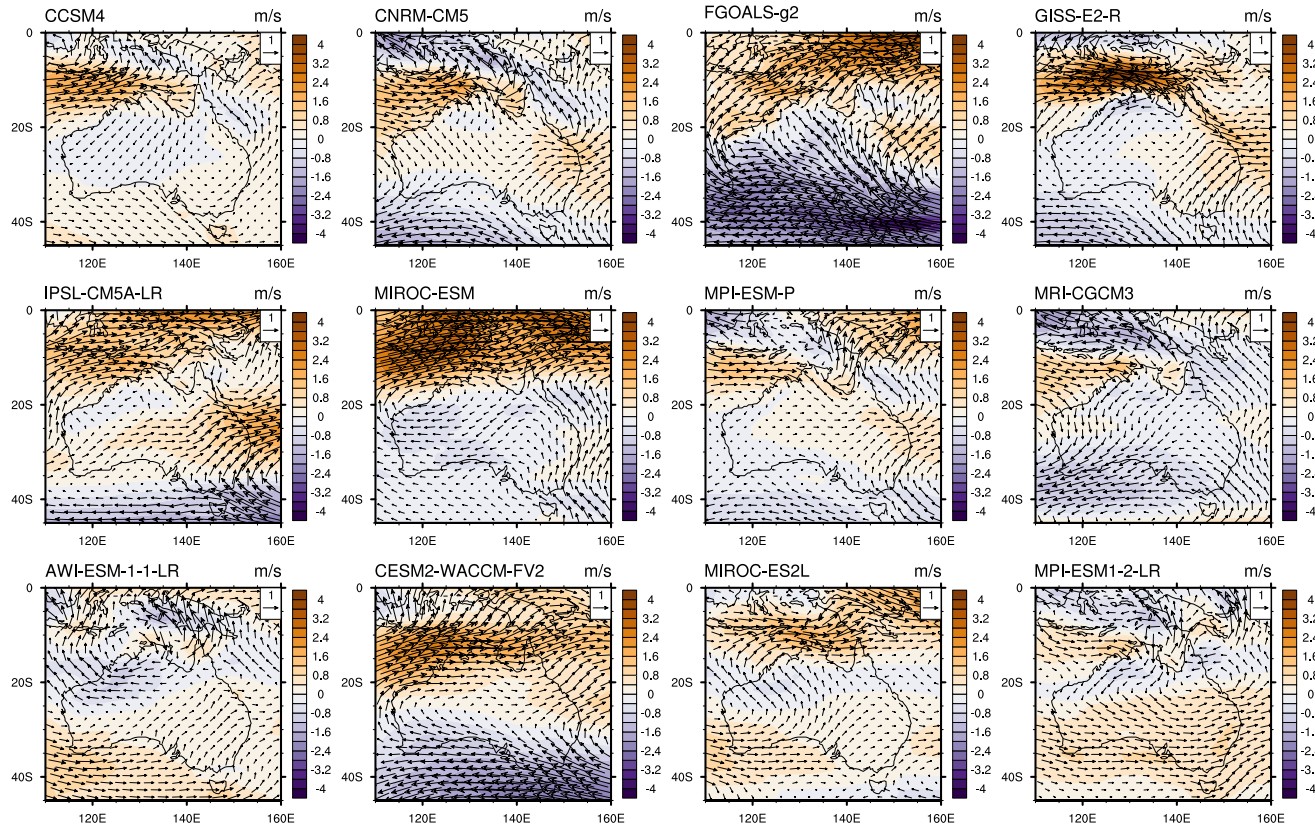

**Figure 7:** Change in mean annual 850 hPa winds (lgm – piControl) for each model and multi-model mean over the Australian domains. Wind change is shown as vectors and zonal (westerly) wind change is shown as colored contours (m/s).


      To further investigate shifts in the position of SH westerly winds or changes in their intensity, the zonal-mean zonal wind is plotted over the SH tropics and midlatitudes. Figure 8 shows the SH (0-70ºS) zonal-mean 850 hPa zonal winds over Australia longitudes (110°E-160°E) in JJA season. All models except CMIP5 CCSM4, MRI-CGCM3 and CMIP6 MPI-ESM1-2-LR models show weakening or no change of SH Australian region mid-latitude westerly strength at the LGM compared to PI. CMIP5 CNRM-CM5, MPI-ESM-P, and CMIP6 MIROC-ES2L models do not show clear evidence of latitudinal changes for maximum mid-latitude zonal wind speed at the LGM relative to PI. A weak equatorward shift in maximum zonal wind speed was simulated in CMIP5 IPSL-CM5A-LR, MIROC-ESM, and CMIP6 MPI-ESM1-2-LR models. In contrast, there seems to be opposite poleward shifts of the maximum zonal winds simulated in CMIP5 CCSM4, FGOALS-g2, GISS-E2-R, MRI-CGCM3 models and CMIP6 AWI-EESM-1-1-LR, CESM2-WACCM-FV2 models.


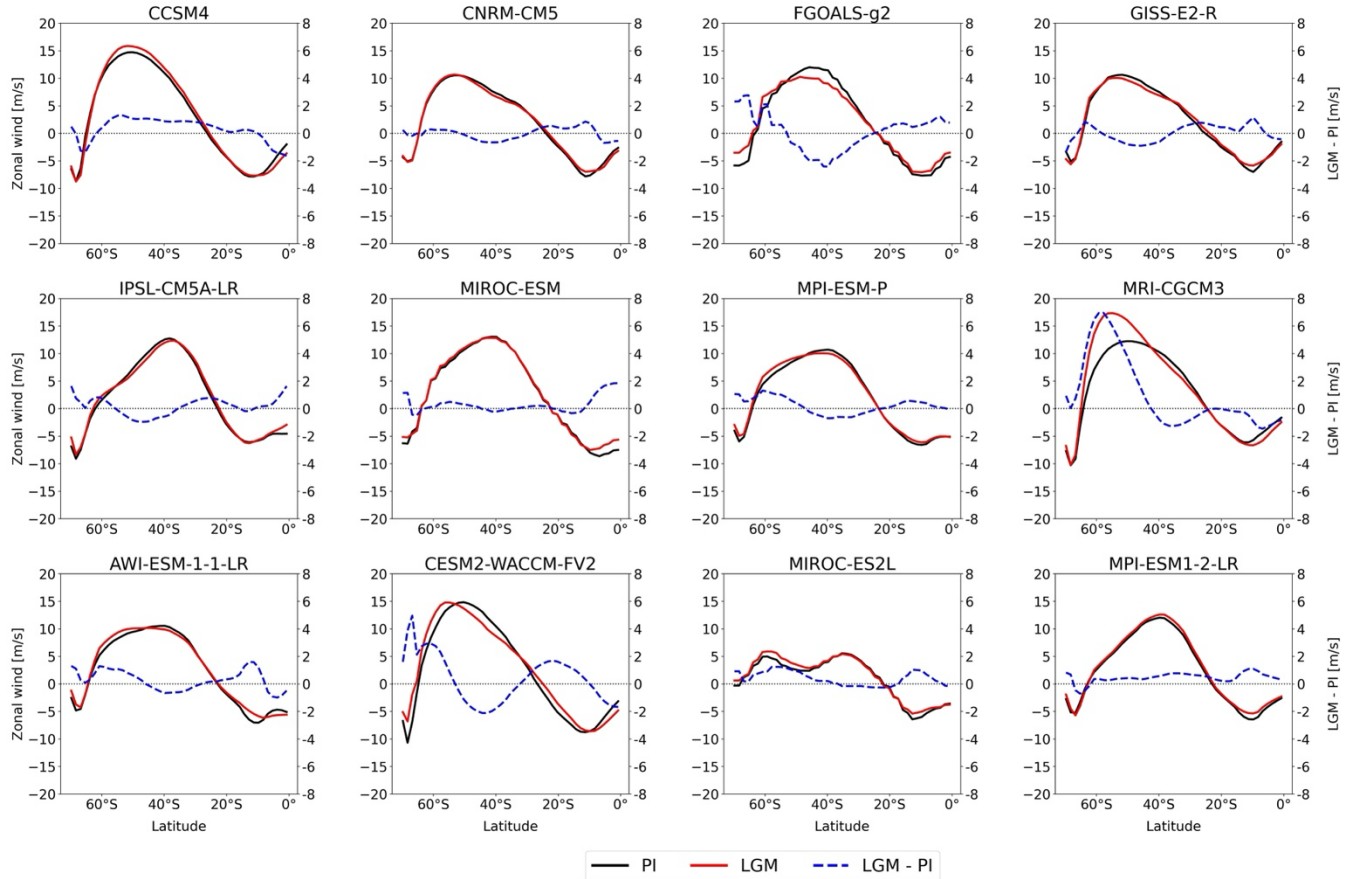

**Figure 8:** JJA zonal-mean zonal wind (m/s) at 850 hPa simulated by individual CMIP5 and CMIP6 models (except INM-CM4-8 model which provided only near-surface winds) averaged over Australian longitudes (defied as 110°E -160°E) for SH (0 to 70ºS). PI in black colour and LGM in red colour, blue dashed line is LGM anomalies (LGM - PI). The LGM - PI zonal wind anomaly (m/s) for each model is displayed on a secondary y-axis. Zonal mean is calculated over all longitudes.

## 3.3 Precipitation

As outlined in Section 2.3, in the present-day climate, regional precipitation variations over Australia are associated with seasonal variations in atmospheric circulation. Austral summer (DJF season) is the period when the Indo-Australian summer monsoon is most active, bringing precipitation to northern Australia and nearby Maritime Continent areas. In austral winter (JJA season), the SH westerlies shift equatorward, bringing precipitation to Southern Australia. This section examines the LGM changes in precipitation and moisture balance over the Australian region relative to PI with a focus on seasonal anomalies. The model agreement is also considered.

The LGM - PI annual average precipitation anomalies over Australia for each model are shown in Figure 9, with
LGM and modern coastlines shown. It is evident that the largest model precipitation anomalies occur over Northern
Australasia, due to the higher absolute precipitation totals in the tropics. Some models (e.g. GISS-E2-R and INM-CM4-8)
simulate increased annual average precipitation over the Sahul shelf region whereas most of the other models show drier
conditions. There is no clear relationship between precipitation anomalies and the extent of the exposed land area over the
Sahul shelf, with some models having small exposed land areas and drying (e.g. CNRM-CM5) and other models having larger
exposed land areas and drying (e.g. FGOALS-g2). A sharp precipitation gradient was simulated along the LGM coastlines in
some models (e.g. MPI-ESM-P, AWI-ESM-1-1-LR and MPI-ESM1-2-LR). Most models agree on an annual mean decreased
precipitation at the LGM averaged over the Australia mainland (see Table 3), however, regional increases in precipitation are
also simulated by most models, indicating that thermodynamic reductions in LGM precipitation due to cooling may be offset
by dynamical increases at regional scales. Individual model precipitation changes in the DJF and JJA seasons are shown in
Supplementary Figures S2 and S3, also displaying model disagreement over tropical regions in DJF but widespread drying in
most models in JJA. Drying in JJA may be due to a combination of thermodynamic and dynamic processes, for example the
poleward displacement of the westerlies in some models, identified in Section 3.2.

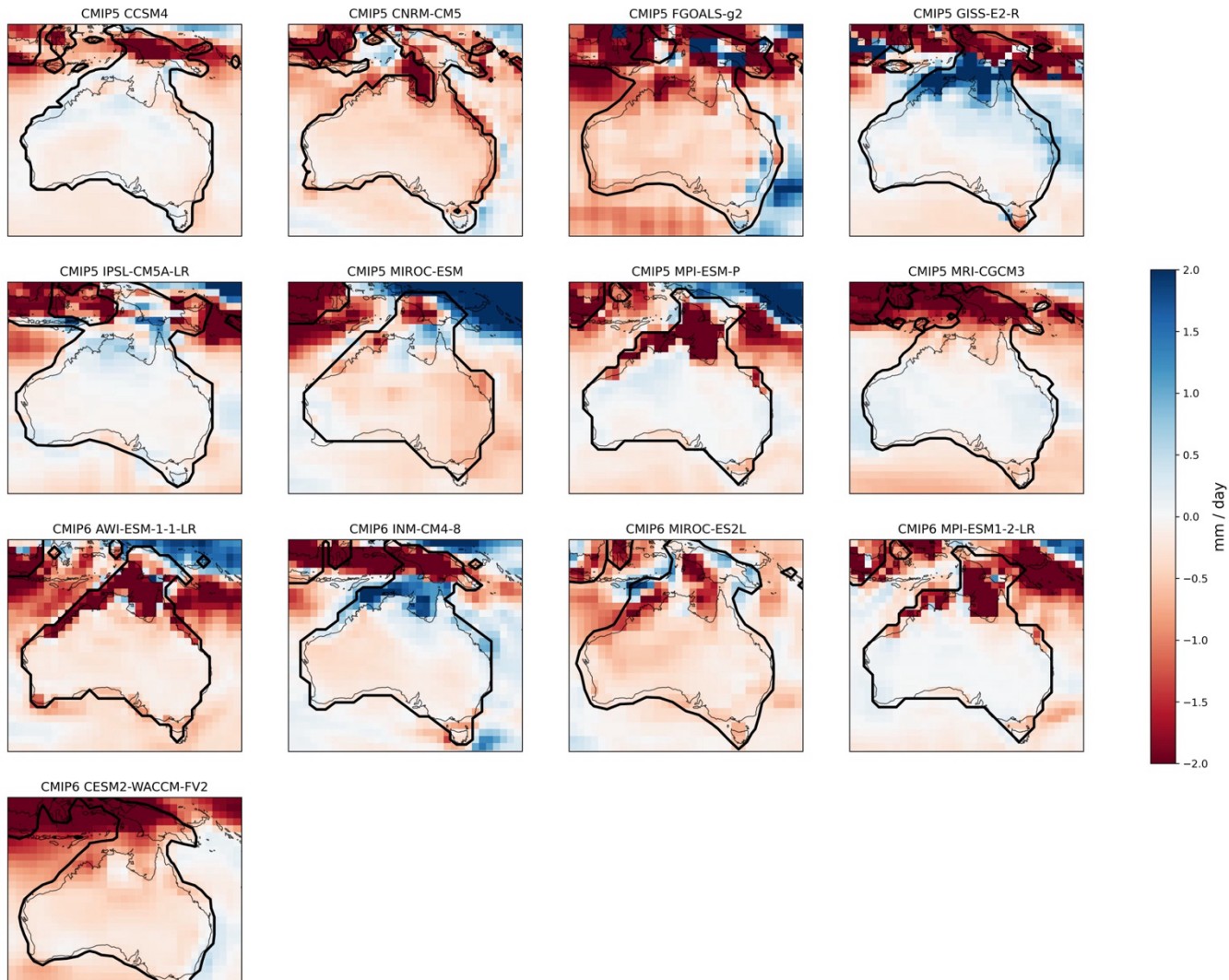


**Figure 9:** LGM – PI mean annual precipitation anomaly (mm/day) simulated by individual CMIP5 and CMIP6 models over the Australian domains. The thick black lines denote the coastlines in LGM prescribed in each model, and the thin black lines denoted the coastlines in piControl. 30% land area fraction was selected for plotting the LGM contours in models.

370         MMM seasonal precipitation anomalies from the combined CMIP5 and CMIP6 model ensemble at the LGM over Australia are shown in Figure 10 and seasonal precipitation changes over specific domains are summarised in Table 3. Stippling denotes regions where less than 70% of ensemble members agree on the sign of the anomaly. In austral summer (DJF) season, an increase in precipitation can be seen over parts of Northern Australasia, particularly the central northern Australian region, while other regions become drier (Figure 10a). Most parts of Australia become drier in MAM, with the strongest drying over

the Sahul shelf and Maritime Continent, indicating an earlier monsoon retreat (Figure 10b).

Austral winter (JJA) season shows the strongest area-average land drying (-0.59 mm/day) over Australia (0-45°S, 110°E-160°E) during the LGM (Figure 10c). There is a strong precipitation reduction over northern Australia and the Maritime Continent in JJA, consistent with previous analysis of PMIP3 LGM simulations (Yan et al., 2018). The south-western corner of Australia shows an average increase in precipitation in JJA but with low model agreement (Figure 10c). There is a strong

increase in precipitation in SON, with high model agreement, over the exposed Sahul shelf and central northern Australia (Figure 10d), as well as along the east coast. This is also consistent with Yan et al. (2018), who identified increased Australian monsoon precipitation in early austral summer (November-December) in PMIP3 LGM simulations.

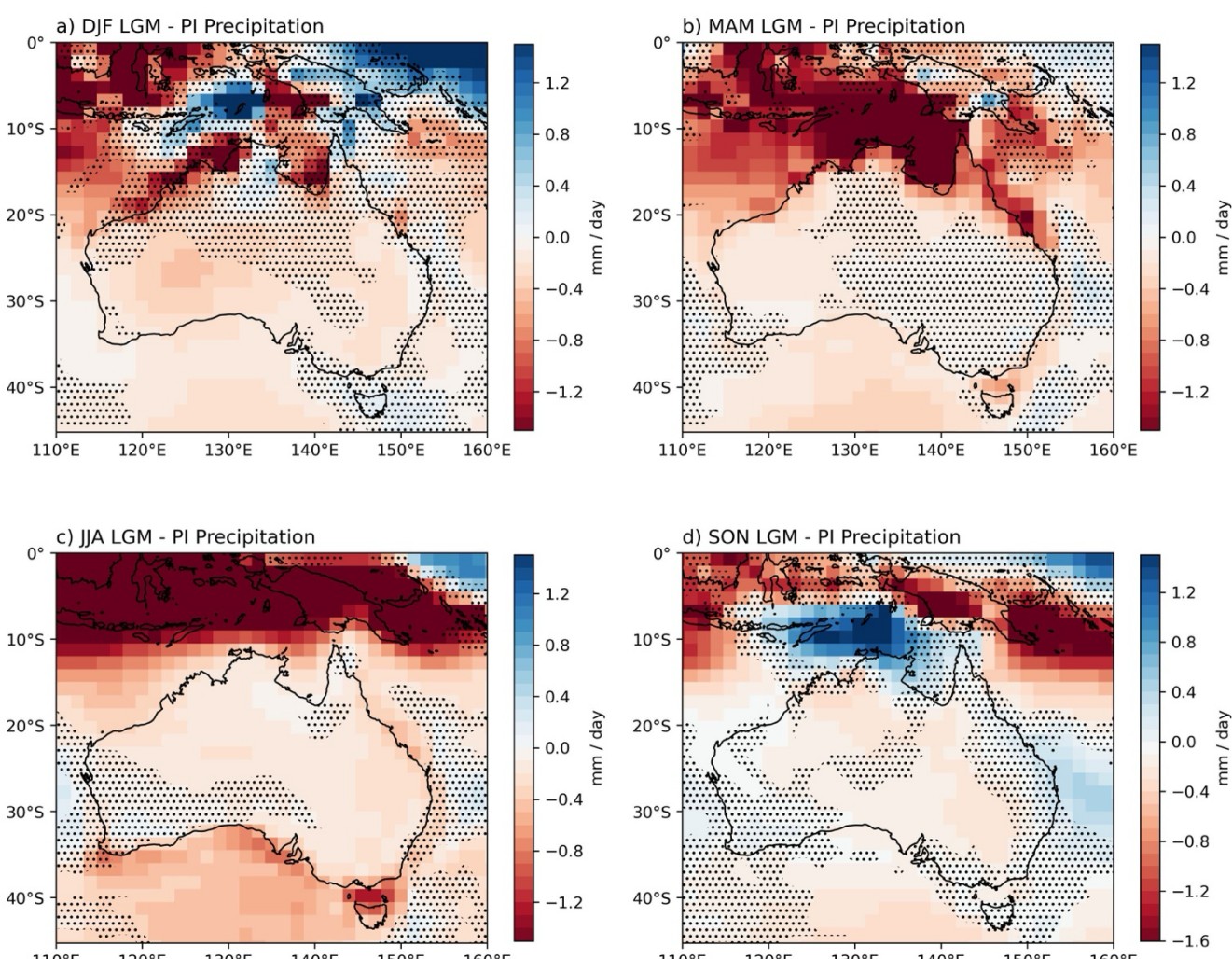

**Figure 10:** MMM seasonal anomalies for LGM – PI precipitation (mm/day) simulated by the ensemble of CMIP5 and CMIP6 models for (a) DJF, (b) MAM, (c) JJA and (d) SON seasons over Australian region. Stippling indicates areas where less than 70% of ensemble members agree on the sign of the anomaly.

### 3.3.1 Precipitation minus evapotranspiration (P-E)

As some proxy records provide information about available moisture or effective precipitation changes at the LGM relative to PI, instead of direct evidence for precipitation changes (e.g. Petherick et al., 2013; Sniderman et al., 2019; Fitzsimmons et al., 2012), we also examine changes in the moisture balance or P-E (precipitation minus evapotranspiration) in the LGM simulations. Cooler conditions at the LGM may lead to reduced evaporation and hence a positive P-E despite reduced precipitation (e.g. Scheff et al., 2017; Kageyama et al., 2021).

Annual mean LGM precipitation and evapotranspiration changes over the Australian regions are shown in Figure 11a and b while net precipitation (P-E) is shown in Figure 11c (seasonal mean evapotranspiration changes are shown in Supplementary Figure S4). There is an annual mean reduction in precipitation during the LGM across the Australian mainland, New Guinea and parts of the Maritime Continent, with an annual average land precipitation change of -0.3 mm/day (based on PI land mask, so excluding coastal areas which are land in the LGM only). The strong precipitation reductions simulated in the expanded land areas in the Northern Australasia region and also the exposed Bass Strait region are associated with decreased evaporation rates at the LGM, so that overall increased LGM P-E was simulated (Figure 11).

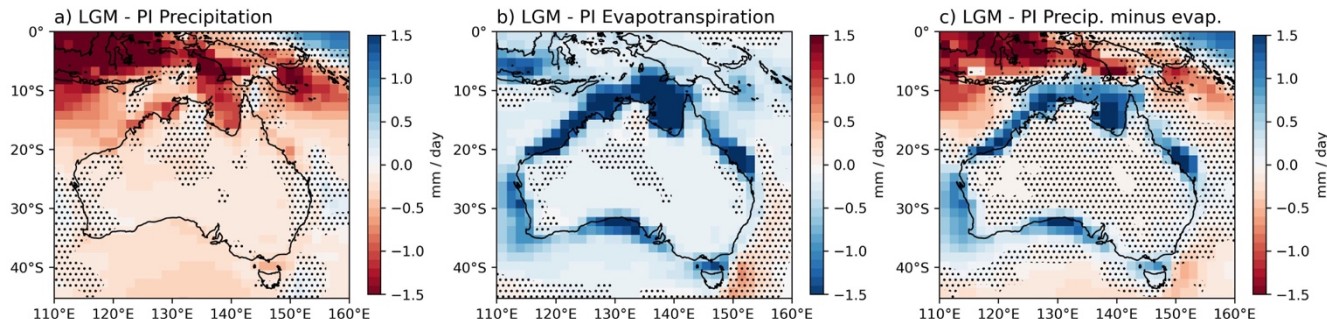

**Figure 11:** LGM – PI mean annual (a) precipitation (b) evapotranspiration and (b) P-E (in mm/day) simulated by the ensemble of CMIP5 and CMIP6 models over Australian region. Stippling indicates areas where less than 70% of ensemble members agree on the sign of the anomaly.

Figure 12 shows the LGM seasonal variations in net precipitation (P-E) over Australia, which can be compared with the seasonal changes in precipitation shown in Figure 10. In comparison to the variable precipitation pattern at the LGM (Figure 10), the P-E pattern (Figure 12) provides more seasonally consistent spatial changes over Australia during the LGM with the largest increase in P-E over the expanded land areas at the LGM. It is also important to note that model disagreement over the sign of the P-E change is large due to the small magnitude of the changes inland from coastal areas.

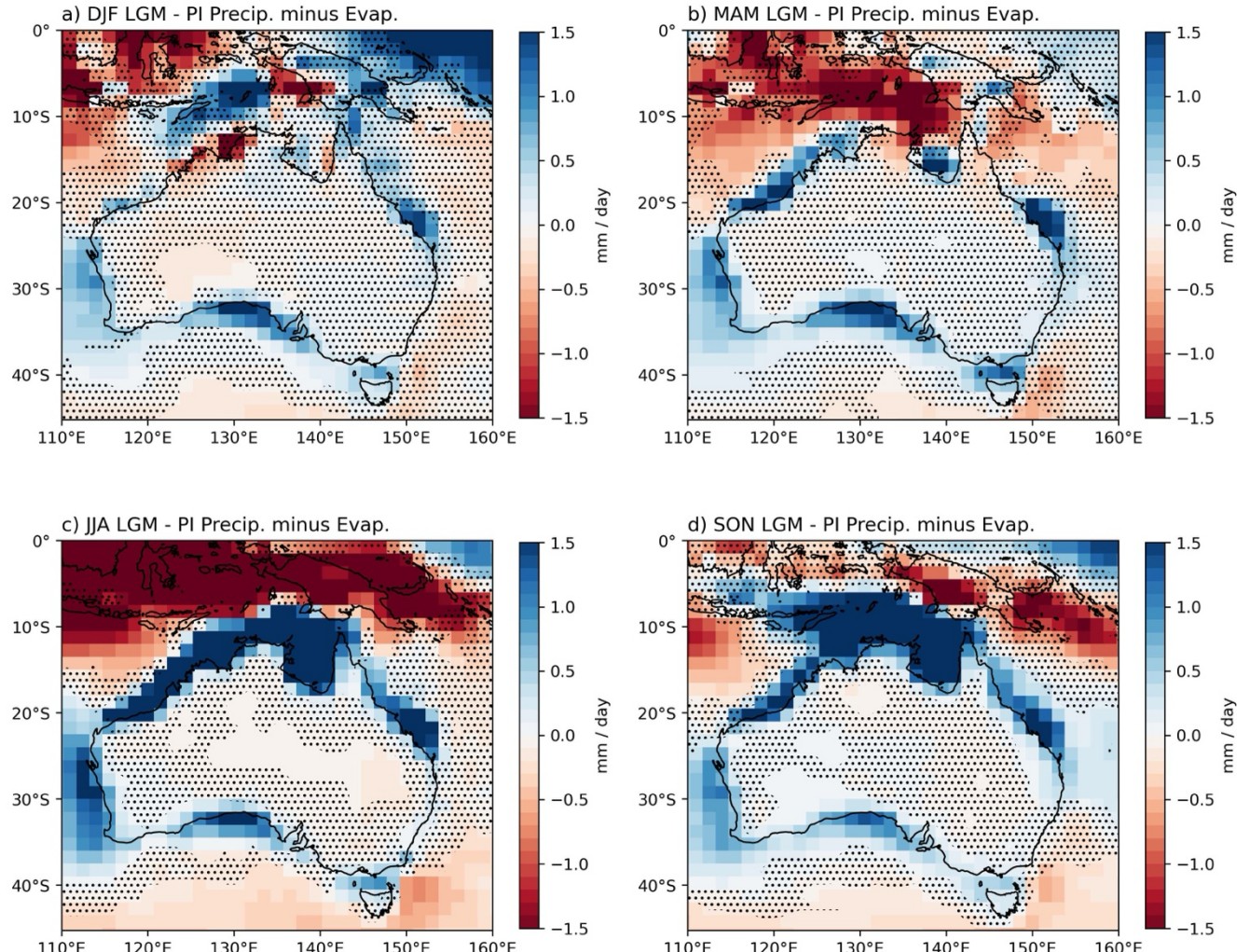

**Figure 12:** MMM seasonal anomalies for LGM – PI precipitation-evapotranspiration (P-E, mm/day) simulated by the ensemble of CMIP5 and CMIP6 models for (a) DJF, (b) MAM, (c) JJA and (d) SON seasons over Australian region. This can be compared with MMM seasonal precipitation shown in Figure 10 and MMM seasonal evapotranspiration shown in Supplementary Figure S4. Stippling indicates areas where less than 70% of ensemble members agree on the sign of the anomaly.

Table 3 shows the average land precipitation and P-E changes at the LGM in individual models over Northern Australasia in DJF and Southern Australia in JJA and the annual mean anomalies over the mainland Australian area south of 10°S, excluding the Maritime Continent. The Australian mainland region experienced an increase in P-E simulated in all models, with a MMM P-E increase of 0.14 mm/day (Table 3). Individual model simulations of annual average LGM P-E changes are shown in Supplementary Figure S5. The simulated P-E pattern show overall good match with the LGM contours in each model, with increased annual P-E along the LGM expanded coastlines.

In the Northern Australasia domain, most models show consistent DJF drier conditions in both average LGM precipitation and P-E changes, although large model disagreements occur (see DJF P-E changes in individual models in Supplementary Figure S6, with less consistent change in response to LGM land expansion). All models except GISS-E2-R simulated more positive P-E anomalies compared to the corresponding LGM precipitation change. In the Southern Australia region, all models agree on reduced JJA area-average precipitation (see individual model simulations in Supplementary Figure S3) and positive P-E anomalies at the LGM, with the exception of FGOALS-g2 model which has a very small negative JJA

P-E anomaly (-0.05 mm/day) (Table 3). Mean P-E changes at the LGM in JJA season for each model over the Australian domain are shown in Supplementary Figure S7, increased P-E is simulated along the expanded southern coastlines in all models. Overall, it is evident that the LGM changes in P-E are less negative (DJF) or positive (JJA and annual average) compared with negative precipitation changes for the MMM and many of the individual models.


**Table 3:** Average land precipitation and P-E LGM anomalies (mm/day) over Northern Australasia (0-20°S, 110°E-160°E) in DJF season. JJA land averages over Southern Australia (20°S-45°S, 110°E-160°E). Annual mean land average changes for modern Australia mainland (10°S-45°S, 110°E-160°E, excluding Maritime Continent) for each CMIP5 and CMIP6 model and MMM. Land is defined by PI land mask.

| Model name | DJF average over Northern Australasia land (mm/day) | | JJA average over Southern Australia land (mm/day) | | Annual average over modern Australia mainland (mm/day) | |
|---|---|---|---|---|---|---|
| | Precipitation | P - E | Precipitation | P - E | Precipitation | P - E |
| AWI-ESM-1-1-LR | -0.85 | -0.64 | -0.25 | 0.06 | -0.38 | 0.00 |
| CCSM4 | -0.48 | -0.24 | -0.20 | 0.03 | -0.06 | 0.18 |
| CESM2-WACCM-FV2 | -1.88 | -1.47 | -0.29 | 0.01 | -0.30 | 0.16 |
| CNRM-CM5 | -0.63 | -0.12 | -0.15 | 0.19 | -0.38 | 0.10 |
| FGOALS-g2 | -0.58 | -0.14 | -0.70 | -0.05 | -0.44 | 0.04 |
| GISS-E2-R | 1.02 | 0.70 | -0.14 | 0.06 | 0.18 | 0.31 |
| INM-CM4-8 | -0.19 | 0.18 | -0.07 | 0.21 | -0.06 | 0.22 |
| IPSL-CM5A-LR | 1.10 | 1.17 | -0.19 | 0.10 | 0.04 | 0.28 |
| MIROC-ES2L | -0.85 | -0.39 | -0.19 | 0.11 | -0.37 | 0.05 |
| MIROC-ESM | 0.53 | 0.74 | -0.26 | 0.10 | -0.27 | 0.11 |
| MPI-ESM-P | -0.83 | -0.20 | -0.03 | 0.10 | -0.12 | 0.05 |
| MPI-ESM1-2-LR | -0.61 | -0.25 | -0.03 | 0.11 | -0.08 | 0.08 |
| MRI-CGCM3 | -1.23 | -0.86 | -0.04 | 0.25 | 0.00 | 0.30 |

| MMM | -0.42 | -0.12 | -0.20 | 0.10 | -0.17 | 0.14 |


### 3.3.2 Drivers of precipitation change

LGM cooling is likely to lead to overall drier conditions due to the thermodynamic response (a colder atmosphere holds less moisture), as seen in the annual mean MMM change (Figure 2). However, the models do not simulate drying over all regions or in all seasons over Australia. For instance, annually wetter conditions are found over the south-west of Australia and parts

of the tropics in some models (Figure 9) and regions of increased MMM precipitation are found over the tropics in DJF and SON (Figures 10). The drying response seen over much of the Australian land area in individual models and the seasonal MMM precipitation changes is consistent with the expected thermodynamic response.

Over the Northern Australasia domain, the seasonal precipitation changes are complex and not clearly linked to temperature changes over either land or ocean. However, changes in temperature gradients between land and ocean may be

important in driving circulation change (see Supplementary Figure S8 for annual surface temperature changes over Northern Australia in each model, plotted over LGM coastlines). The exposure of the Sahul shelf appears to drive changes in moisture transport, with increased onshore westerlies (Figure 6) driving convergence upstream of the LGM coastline, producing increased precipitation to the northwest of Australia over a small region in DJF (Figure 10a) and a larger region in SON (Figure 10d). The disagreement in simulated austral summer precipitation between models (Supplementary Figure S2) can also be

linked to different changes in circulation in the models (not shown). Offshore wind anomalies and increased wind divergence in MAM and JJA produces widespread drying over the entire Northern Australasia region. Further analysis of moisture convergence fields is required to confirm this mechanism, but it is broadly consistent with the findings of Yan et al. (2018) based on PMIP3 LGM simulations.

LGM precipitation changes over Southern Australia appear to be dominated by thermodynamic drying, with

additional contributions from dynamical processes including shifts in the position or intensity of the SH mid-latitude westerlies. While area average precipitation changes over the Southern Australia domain are negative in JJA, the majority of models simulate increased precipitation over the south-west corner of Southern Australia (see Supplementary Figure S3). This region is also climatically distinct in the modern climate, experiencing higher winter rainfall than surrounding areas, with a recent strong drying trend (Hope et al. 2010).

We explored the relationship between the JJA LGM change in precipitation averaged over south-west Australia (30ºS-35ºS, 115ºE-120ºE) and the change in strength of the westerly winds over the same latitude range, but with a wider longitude range for the entire Southern Australian region (110°E-155°E) as well as the latitudinal shift of maximum westerly wind strength. The corresponding scatter plots are shown in Supplementary Figure S9. The Pearson correlation coefficient between the westerly strength and precipitation is $r = 0.537$, and the corresponding p-value is 0.089, while the correlation between shifts

in the latitudes of maximum westerly winds and precipitation is $r = 0.586$, with a p-value of 0.058, indicating a slightly stronger

correlation but still not significant at the 95% confidence level. Sime et al. (2013) found that there is no clear relationship between the displacements of westerlies and enhanced precipitation at the LGM based on PMIP models, which is consistent with our findings. However, the positive anomalies in the south-west indicate a dynamical driver of precipitation increase is stronger than thermodynamic drying in this region.

## 4 Discussion and Conclusions

This study has evaluated the temperature, precipitation and wind changes over the Australian region at the LGM compared with pre-industrial conditions from CMIP5 and CMIP6 model simulations. We now summarise the key finding from the model simulations and briefly compare these with published proxy records of LGM temperature, precipitation and atmospheric circulation from this region, introduced in Section 1.

### 4.1 Temperature

The LGM ensemble of thirteen CMIP5 and CMIP6 models simulates a global annual cooling of surface air temperature of -4.5 ºC compared to PI conditions, within the range proposed in other modelling studies (e.g. Annan et al., 2022; Tierney et al., 2020a). Regionally, the models simulate cooling of annual surface air temperatures in the range of -1.8 ºC to -5.5 ºC (MMM = -2.9 ºC) in the Australian domain (0-45°S, 110°E-160°E), with larger cooling over land and particularly over coastal regions with expanded land area at the LGM. On seasonal time scales, cooling is slightly larger in MAM and JJA over this region. Our model results for temperature change generally agree well with available proxy records and previous modelling studies although there are some LGM proxy records that imply greater LGM cooling than the models. For example, Miller et al. (1997) and Hope (2009) reconstructed greater cooling over land than was simulated by the models, with Miller et al. (1997) finding cooling of 9 ºC in inland Australia and Hope (2009) finding cooling of 4-6 ºC in the New Guinea highlands. This may be due to uncertainties in the proxy reconstruction. Alternatively, the models may not be simulating the extent of cooling over land in the LGM simulations. In the case of proxy records over high topography (e.g. Hope, 2009), the models do not resolve the details of the topography and associated cooling.

### 4.2 Precipitation

The models simulate an overall reduced precipitation over the Australian region, with negative annual average anomalies in the majority of models and the MMM. Seasonal MMM precipitation anomalies (Figure 10) indicate widespread, slightly lower precipitation than today, except for slightly higher precipitation during DJF, in the north-eastern central Northern Australia, northern Cape York, and all of Tasmania except the west coast; during JJA in SW western Australia; and during SON in the subtropical eastern margin of the continent. Examination of the P-E patterns (Figure 12) shows most of the continent experienced reduced precipitation but positive P-E anomalies, implying that most regions at the LGM would have experienced little change in moisture availability, or slightly wetter conditions, particularly on seasonal time scales.

Comparison of available proxy records for moisture availability with model simulations at a regional scale indicates some agreement. Models simulate summer (DJF) reduced moisture availability (P-E) over the north-west region corresponding to the modern coastline but inland from the LGM coastline, which may be consistent with drier summer monsoon conditions at Girraween Lagoon (Rowe et al., 2021) and Ball Gown Cave (Denniston et al., 2013) although with low model agreement. Increased annual mean and cool season LGM moisture availability is simulated in coastal regions of Southern Australia, which may be consistent with higher moisture availability based on lake and fluvial records (e.g. Kemp et al., 2017; Hesse et al., 2018). Further inland, there is a weakly positive or near zero LGM moisture balance anomaly over the regions corresponding to Lake Frome (Cohen et al., 2015) and Mairs Cave (Treble et al., 2017), while these records indicate wetter conditions. In addition, the large inter-model disagreement (stippled areas in Figures 11 and 12) suggests that at least some models are not realistically simulating the LGM hydroclimate response in this region. The reasons for this are beyond the scope of the present study, but likely reflect biases in model convection, land surface interactions or other relevant processes.

## 4.3 Winds

Similar to many previous studies, we did not find a consistent equatorward or poleward shift in the SH mid-latitude westerly winds at the LGM over the Australian longitude range or a wider SH zonal average (not shown) based on the available CMIP5 and CMIP6 model simulations. Over the south-west corner of Southern Australia, increased LGM precipitation in austral winter (JJA) was more likely in models with strengthened SH westerlies over the surrounding domain, although the correlation was not significant. However, we did not identify a consistent shift in maximum SH mid-latitude zonal winds in the LGM simulations. Future work could calculate the SST front latitude in models as a new method of quantifying meridional shifts in SH surface westerly winds at the LGM, as proposed by Gray et al. (2023). Moreover, the consideration of impacts from sea ice variability at the LGM to SH westerly changes should be also taken into account in future research (Chavaillaz et al., 2013).

## 4.4 Conclusions

This research presents an initial evaluation and analysis of climate conditions (temperature, precipitation, wind) over the Australian domain (including adjacent regions of Indonesia and New Guinea) at the LGM using an ensemble of eight CMIP5 and five CMIP6 models with available PMIP LGM simulations. The results offer insights into regional Australian climate variations during the LGM, and the inter-connections between climate variables in different seasons. Our model results show widespread cooling in the annual mean and all seasons, with a magnitude that is generally consistent with available proxy records except in central Australia and high elevations in New Guinea, where models show less cooling than proxy records.

The models disagree on changes in precipitation over Northern Australasia in austral summer, with a complex multi-model mean pattern of precipitation change. The lack of model agreement indicates that each model responds differently to the specified changes in boundary conditions at the LGM, producing different changes in circulation and therefore different dynamical components of the precipitation response. In addition, different boundary conditions configurations in models (e.g.

land-sea mask) may also play a role in varying the precipitation response. This implies that caution is required when comparing proxy records with simulations from a single model or small group of models in this tropical region. Given the large uncertainty (including in the sign of change) in both models and proxies, it is not currently possible to make robust conclusions about whether it was wetter or drier over Northern Australia at the LGM based on the models and proxy evidence.

In Southern Australia, the changes in precipitation indicate widespread drying over land in all seasons, except for the southwest of the continent. The changes in the position or intensity of the SH westerlies are not consistent between models. Given the large changes in global and regional temperature in the LGM simulations, with similar patterns of cooling in all models, it is noteworthy that the models do not simulate a consistent shift in this important component of the regional circulation, suggesting that shifts in the position of the westerlies cannot be assumed as a simple response to large-scale climate cooling (or warming).

Further analysis is required to better understand the divergent model responses to LGM boundary conditions in both the tropics and mid-latitudes in the Australian region. This analysis may also provide guidance for understanding model responses in a future warming climate. The sensitivity of model changes in tropical precipitation to the representation of land area change requires further investigation, and future PMIP LGM simulations should use similar land masks over Sahul if possible. In addition, a larger ensemble of LGM simulations would be beneficial to allow selection of models with good performance over the region of interest. Comparison with well constrained proxy records of past hydroclimate changes would also provide a useful target for identifying more realistic model simulations. Such comparison will need to consider the moisture balance rather than focusing only on precipitation, and will need to account for sensitivity of some proxies to changes in $CO_2$ levels.

**Code and data availability**

The CMIP5 and CMIP6 model data used in this study are available from the Earth System Grid Federation.

**Supplement**

The Supplement related to this article is available online at:

**Author contributions**

YD and JRB designed the study. YD carried out the data analysis and led the writing of the manuscript. JRB and JMKS provided comments on the results and contributed to the writing of the manuscript.

**Competing interests**

The authors declare that they have no conflict of interest.

**Acknowledgments**

Josephine R. Brown and Yanxuan Du acknowledge support from the Australian Research Council Centre of Excellence for Climate Extremes (CE170100023). We acknowledge the World Climate Research Programme, which, through its Working Group on Coupled Modelling, coordinated and promoted CMIP6. We thank the climate modelling groups for producing and making available their model output, the Earth System Grid Federation (ESGF) for archiving the data and providing access, 565 and the multiple funding agencies who support CMIP6 and ESGF. CMIP5 and CMIP6 model outputs were made available with the assistance of resources from the National Computational Infrastructure (NCI), which is supported by the Australian government.

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
