# Peer review of "Last Glacial Maximum Climate and Atmospheric Circulation over the Australian Region from Climate Models"

_Climate of the Past, 2023_

## Author Comment (AC1)

The knowledge of the characteristics and mechanism for the climate change over Australian regions in LGM is still not enough. This study investigated the climate changes at the LGM over the Australian region, in terms of temperature, precipitation, moisture balance and wind, based on the output from PMIP3 and PMIP4 simulations. The work might contribute to our understanding of the hydrological change of Australia in ice ages. The following are my comments and reviews for the authors' consideration.

***Comments:***

1. The uncertainties of model simulations and reconstructions are important information for model-data comparison. It would be better to evaluate the model consistence since there was large model spread. Specifically, the authors could further provide the percentage of model ensembles consistent on of the signal of their multiple model ensembles mean value.

[Response]: Thanks. Stippling will be added to all MMM figures (Figure 3, 5, 8, 9, 10) to show model agreement. All other figures show individual models which allows model consistency to be evaluated by the reader.

For the reconstruction, the background information of proxy used here and their uncertainty could be listed in a table. The information of the LGM climate getting wetter or drier and in which parts of Australia based on proxies is still not clear, even though the authors cited others' work in line 459-463. It would be easier to read and make comparison if were there reconstructed data mapped on the plots of model results.

[Response]: Thank you for your suggestion. We emphasize that this study is not concerned specifically with detailed data-model comparison of LGM hydroclimate in Australia. Our principal reason for avoiding a quantitative data-model comparison with proxy climate reconstructions of the Australian LGM is that the Australian proxy-based LGM palaeoclimate literature typically has provided qualitative reconstructions, ('drier', 'much drier', 'somewhat drier') and that, where quantitative reconstructions have been provided, they have uniformly been based on comparison of LGM plant distributions with modern plant distributions, or comparison of the LGM occurrence of mobile sand dunes with their modern distribution, respectively. These comparisons, however, have ignored the plant physiological effects of low atmospheric $CO_2$, which is increasingly recognised as a problem that should not be ignored (Scheff et al., 2017; Prentice et al., 2022). In brief, C3 plants perceive an LGM world with low (180 ppm) $CO_2$, as much 'drier' than today, so that model simulations with dynamic vegetation typically show widespread forest reduction, even when holding temperature and precipitation at modern values. Moreover, it is increasingly suspected (e.g. Scheff et al., 2017; Roderick et al., 2015) that greater LGM dustiness and dune-mobilisation are secondary effects from reduced plant productivity, via landscape destabilisation.

For these reasons, we believe it is misleading to describe or list the published LGM hydroclimatic reconstructions in detail. We believe it will be clearer, to explain in summary form the recently-recognised problems associated with understanding LGM hydroclimate, as described above; and treat the published reconstructions collectively, at the level of a literature, rather than as individual reconstructions.

1. Usually, modeling community use surface air temperature (SAT) instead of surface temperature (ts) to investigate the temperature change, and to explain the related change of circulation and/or precipitation.

[Response]: Thank you for that. We will redo all temperature analysis for surface air temperature (tas).

2. The climate proxy from 28 to 18 ka is compared with the LGM simulations at 21ka (Line 55-56). This may also contribute to the model-data inconsistence considering the extended date of proxy. For example, the variability of climate proxy during 28 and 18 ka may switch between drier or wetter condition than pre-industry and thus make the complexity of model-data comparison. This point could discussed further when necessary

[Response]: Thank you for the suggestion. We have checked the dates of the proxy records, this range can be narrowed to 24-18 ka, which is consistent with many studies of LGM climate (e.g. Clark et al., 2009).

3. As pointed out by the authors, the sst gradients and related circulation change could explain the precipitation change (Line 309-311). Thus the analysis of sst change (which roughly equals to the ts value over ocean) and model-data comparison of sst could improve the knowledge of the LGM climate change over Australia.

[Response]: Thank you for the suggestion. We will add a Supplementary Figure showing the SST change over a larger domain extending across the Pacific and Indian Oceans to allow discussion of SST changes linked with Australian precipitation change. Further analysis of the dynamical links between SST and precipitation are beyond the scope of the current study.

***Line by line comments***

Line 42. "Many regions", could be pointed in details.

[Response]: This sentence will be rewritten to provide more detail of vegetation changes based on proxies and biome models, with citation of Prentice et al. (2011) as the appropriate source: "There was a large reduction in area covered by boreal and temperate forests in northern mid- to high latitudes, expanded lowland tundra in Eurasia, expansion of savanna and grasslands at the margin of Amazon tropical forests and replacement of some areas of tropical forest in Africa, China and Southeast Asia with savanna, woodland and grassland (Prentice et al., 2011)."

Reference:

Prentice, I. C., Harrison, S. P., & Bartlein, P. J. (2011). Global vegetation and terrestrial carbon cycle changes after the last ice age. New Phytologist, 189(4), 988–998. https://doi.org/10.1111/j.1469-8137.2010.03620.x.

Line 92-100. The reconstructed evidence of moisture or hydrocliamte could be compared with model simulations in the section of discussions.

[Response]: Thank you for your suggestion. We already have some discussion in Section 4.2 but will expand this.

Line 155-157. There were three different ice sheet reconstructions. Thus it's necessary to clarify the information of ice sheet configuration of the four models from PMIP4.

[Response]: Thank you for your comment. A new Table will be added which gives information of ice-sheet reconstructions for individual models. The PMIP3 models used PMIP3 ice-sheet configurations and the four PMIP4 models used in this study all used the "ICE-6G_C" ice-sheet reconstruction (differences between the two LGM ice-sheet reconstructions in Kageyama et al., 2017), this will be discussed in Section 2.1.

Table 2. In term of vegetation of PMIP4, were there any model using the dynamic vegetation? Please check and make it clear.

[Response]: Only PMIP4 AWI-ESM-1-1-LR model is using dynamic vegetation. This will be noted in the text in Section 2.1.

Line 180-182. Usually models use the last 100 years, instead of the first 100 years, to do analysis. Were there any big differences between those two choices?

[Response]: Thank you for your comment and sorry for the confusion with the wording of the sentence. We will clarify why we choose the first 100 years of the model run with the new table described above.

Most climate models used in this study only have 100-year length of simulation based on the number of years of data available on the ESGF (Earth System Grid Federation). According to Kageyama et al. (2017), the models have been spun up until equilibrium following the PMIP protocols (refer to Kageyama et al. (2017) for details of the spin-up). At least 100-year data from the equilibrium part of the simulation is required to store on the ESGF (Kageyama et al., 2017). Therefore, the data stored on the ESGF has already been in equilibrium and it does not matter anymore whether it is the first or the last 100 years. There will be no significant differences.

Line 213-214. The difference between the analyses in the paper with Kageyama et al. may lies in the choice of ts, instead of SAT. Please check.

[Response]: Thank you for the comment. Yes Kageyama et al. (2021) used surface air temperature instead of ts. We will change all of our temperature analysis to surface air temperature (tas) to allow a clearer comparison between our study and previous work.

---

## Author Comment (AC2)

**Reviewer#2:**

**General comments**

In this study by Du et al., the authors propose a modelling intercomparison study of the simulated surface temperature, precipitation, moisture balance and winds at the LGM, focusing on the Australian region.

Such a study would fit well within the scope of Climate of the Past. The case study of Australia is interesting for several reasons such as the impact of sea level change on coastlines, the specific location (SH, in proximity to the Maritime Continent and Southern Ocean), and the overall disagreement of models and proxies especially concerning the latitudinal shift and variation of strength of the westerly winds, which seem both poorly represented in PMIP models (Chavaillaz et al., 2013) and poorly constrained by conflicting paleodata records (Kohfeld et al., 2013). The manuscript is also clear and an easy read.

However, as such (and as is often the case with intercomparison studies), the paper reads as very descriptive and superficial, so we are struggling to learn something new. Knowledge gaps, uncertainties in both model and data and processes/mechanisms are either barely mentioned or simply not highlighted enough so when we reach the end, the impression is underwhelming.

I am providing more concrete illustrations as to what could be improved and how with points 1-3. I also have suggestions related to methodology in points 4-8. So I would like to recommend a number of improvements before publication, in the hope of helping give this study more weight.

1. Please elaborate on the reasons why a case study of the past climatic changes over Australia is interesting. It would be great to mention climate processes that are key in this region. An example since dust transport is mentioned (L93): does this aridity have the potential to significantly enhance iron fertilisation in the ocean?

[Response]: Thank you so much for your comments and suggestions. We will add more discussion of why case study is interesting with reference to key climate processes in the region. While the issue of dust transport and iron fertilisation is beyond the scope of the present study, we will include discussion of the major changes in Southern Hemisphere climate and circulation which may have influenced Australia during the LGM, e.g., changes to the SST gradients in the tropical Pacific and Indian Ocean, and related changes in the Walker circulation and Hadley Cell, changes to the position or intensity of mid-latitude westerlies and changes in the strength of the Australian-Maritime Continent monsoon.

2. Consider adding one or two last sentences to the abstract and a paragraph in the discussion/conclusion to give the reader a broader perspective and hindsight on what we have learned and how significant are these new findings. A few questions to help brainstorm: So what? In the basis of the existing literature and these new findings, have we achieved a better understanding of the processes which influenced the past Australian regional climate? If not, what are we missing? Do we understand the model response to LGM forcings? What does it entail?

[Response]: Thank you for your suggestions. We will expand the Abstract and Discussion to include more discussion of the significance of the findings.

3. Please underline the knowledge gaps in the introduction. As such, the introduction is very descriptive (not impactful), with a structure (global changes / changes in different regions of Australia) that doesn't help guide the reader very logically towards understanding the knowledge gaps, their importance, the scientific question and the methodology used in this study. If the authors would like to keep this regional description structure, then it would be welcome to also point out the contrasts between these different regions, and also with the global climate, with a few short sentences to conclude this subsection.

[Response]: Thanks. The structure of the Introduction follows many previous proxy-based studies examining regional changes in Australian climate during this period (e.g. Reeves et al., 2013b; Fitzsimmons et al., 2013; Petherick et al., 2013; etc.). As Australia spans several distinct climate zones, we thought it would be logical to structure the Introduction around these distinct climate zones. We will add several sentences comparing these with each other and with global climate.

Still, I feel like the sometimes conflicting proxy records and the 'uncertainty about the drivers of the LGM climate changes' (L74-75), the 'ambiguous results' of models (L117) should be arguments brought to the reader's attention in a more convincing order to justify the need of this particular intercomparison study and its methods. For exemple, it is not clear what is the advantage of using an intercomparison method, nor for which reasons modellers simulate the LGM period (L102-107). It is also not clear why the authors chose to examine these three specific climate variables. I believe there are ways to reinforce the visibility of the scientific reasoning behind this approach.

[Response]: Thank you for the comments. We note that this approach is widely used in numerous papers using PMIP LGM simulations. We will more clearly explain the motivation for simulating the LGM and for using multiple climate models in Section 2. We will also explain why the particular climate variables were chosen.

4. Only PMIP4 outputs available on the ESGF were included. This is a bit of a shame because it limits comparison with the Kageyama et al. (2021) results, and the model ensemble size (and therefore the robustness of the results). CESM2 is also excluded for very good reasons (L143-144), but I believe the authors have found the source of the exaggerated cooling in a cloud microphysics parameterization and run a corrected simulation. This is of course up to the authors, but they could consider contacting both the Kageyama et al. (2021) and Zhu et al. (2021) authors to request the model outputs. This would also enable the authors to compare individual model versions (CMIP5 vs CMIP6) and discuss potential improvements between the two generations. For now, only MPI-ESM is in the two ensembles.

[Response]: Thanks for your comments. We decided to make use of only those models which are publicly available via the ESGF to ensure our results could be easily reproduced. We follow the standard approach in CMIP climate model studies which typically only use publicly archived simulations. Many other PMIP-based studies have also used this approach, e.g. a recent paper on LGM ITCZ changes from PMIP3/4 models used a similar set of models (Wang et al., 2023).

Regarding CESM2 models, we had some trouble linking the available CESM2 model simulations on ESGF with documentation and relevant publications. We will now include the CESM2-WACCM model as we now understand this model does not have an unrealistic climate sensitivity.

The arge-scale comparison between CMIP5 and CMIP6 models was provided in Kageyama et al. (2021) so we do not need to repeat this analysis. We do not assume that the two generations of a particular climate model (e.g. MPI-ESM) will produce a similar simulation of LGM climate, and we do not wish to focus on comparing generations of specific models – this is typically done by the relevant modelling groups.

The focus of the present study is to summarise the simulation of LGM climate of Australia from available PMIP3 and PMIP4 models. This information is of interest to the Australian palaeoclimate community and will provide a basis for comparison with numerous proxy-based reconstructions of standard climate variables such as surface temperature and precipitation, as outlined in the Introduction.

Reference:

Wang, T., Wang, N., & Jiang, D. (2023). Last glacial maximum ITCZ changes from PMIP3/4 simulations. Journal of Geophysical Research: Atmospheres, 128, e2022JD038103. https:// doi. org/10.1029/2022JD038103

5. The authors mentions that all model outputs are regridded (L170), and it seems that all the following analysis use these regridded outputs. Of course, this is necessary to compute and plot the multimodel mean, but I am wondering whether it would be worth extending the use of the model native grids when plotting the individual maps and wind profiles. Is the latitudinal shift of westerly winds affected by the model resolution? Are there differences in the land-sea mask of each model which could impact the simulated temperature and precipitation patterns? Please consider plotting the PI or LGM land-sea mask (e.g. as a grey or dashed contour) on maps.

[Response]: Thanks for the suggestion. We have recalculated boundary lines between SH easterlies and westerlies on the native model grids (Table 4), however, we decided to remove it from the paper. This is because the results differ quite a lot on the different grids, and we think they are not robust enough to be included. The whole Section 3.2 will be then rewritten, including adding some new figures focused on the large-scale shift in winds.

Regarding the influences from the land-sea masks in different models, we will plot the LGM land masks for individual models in the Supplement document to show the potential impacts from the different land fractions configured in each model.

6. On maps, the authors should also represent the multimodel agreement significance as hashes (when >90%) and the proxy data as scatter points, whenever quantitative reconstructions can be found. I consider important that the reader is able to compare visually the performance of the models with the available proxy data, especially as the authors conclude that there is a general good agreement between model and data.

[Response]: Thanks. We will add stippling for model agreement. As noted for reviewer 1, we have provided already that the sign of the change from proxy records is uncertain, especially when taking into account the $CO_2$ effect on vegetation records, and therefore we refer to the literature but don't include any proxy records in our plots.

7. The authors mention using the first 100 years of simulation for the analysis. Why not the last 100 years? Please check that the simulations are in equilibrium, e.g. by computing the drifts.

[Response]: Thanks. Same as for Reviewer 1, most climate models used in this study only have 100-year length of simulation based on the number of years of data available on the ESGF (Earth System Grid Federation). According to Kageyama et al. (2017), the models have been spun up until equilibrium following the PMIP protocols (refer to Kageyama et al. (2017) for details of the spin-up). At least 100-year data from the equilibrium part of the simulation is required to store on the ESGF (Kageyama et al., 2017). Therefore, the data stored on the ESGF has already been in equilibrium and it does not matter anymore whether it is the first or the last 100 years. There will be no significant differences.

8. The paper would deserve more quantifications. An example is in L273: 'Some models show weakening and other model show strengthening but there are other instances (e.g. L372). It would be great to provide precise figures, e.g. phrasings like '5 models out of 12 show a weakening of at least 20%...'

[Response]: Thank you so much. We will change the wording to quantify model agreement and specify the magnitude of changes as suggested.

**Specific comments**

L2, L12 and L13: 'changes at the LGM', 'to cool by 2.6 at the LGM' and ,'decreased'. Unlike in the rest of the paper, it is sometimes unclear in the abstract that we are mentioning changes relative to the PI. This has to be indicated in some way (e.g. LGM-PI anomaly, with respect to PI…) or else the verbs are inconsistent with the time direction.

[Response]: Thanks. We will change the wording here to avoid confusion.

L13-14: Why are the changes in temperature and precipitation indicated over two different defined regions?

[Response]: Thanks for asking. The calculation for MMM precipitation changes was over a smaller domain aiming to exclude the land areas between 0 to 10°S to allow comparison with previous studies focused on northern Australian rainfall.

L18-19: I find the sentence explaining the potential reasons for model-data disagreement to be rather vague and could be reformulated.

[Response]: We will rewrite it to improve clarity.

L23: The time window proposed for the LGM is unusual and only justified later in the text.

[Response]: Thanks. We will change it to "22-18 thousand years ago (ka)".

L31: Why such a gap between the Annan et al. (2022) and the Tierney et al. (2020a) estimates?

[Response]: This disagreement is due to the "choice of prior", i.e. the particular climate model used in the Tierney et al. (2020) study, as discussed by Annan et al. (2022). We will briefly note this.

Fig 1 and 2: These figures do not bring a lot of information to the table. The authors could consider combining them, combining Fig 1 with e.g. Fig 3 (the land-sea mask could be indicated on another map with a grey contour), or enriching them with more information (e.g. the Sunda and Sahul shelves mentioned in L34 could be annotated on the map to show the reader their location). As for Fig 2, consider using a different style (than the red contour, e.g. hashes) for the southwest box.

[Response]: Thanks for the suggestion. We decided to combine Fig 1 and Fig 2 with the LGM land mask for CCSM4 model as background shading. The southwest box will be plotted using a different style.

L85: The authors could elaborate on the reasons why they used the word 'possibly'.

[Response]: According to Kohfeld et al. (2013), "If a single cause related to the southern westerlies is sought for all the evidence presented, then an equatorward displacement or strengthening of the winds would be consistent with the largest proportion of the observations." This was summarised with the word "possibly", but we will rewrite this as "is consistent with" to better summarise the findings of the Kohfeld et al. (2013) study.

L110: 'not fundamentally different' could be true for the variables examined in the study and not others. Please check if this is the case for all variables (including ocean circulation).

[Response]: The global study of Kageyama et al. (2021) has already addressed the comparison between PMIP3/CMIP5 and PMIP4/CMIP6 models. While individual models will differ for any variable considered, the sample spread of PMIP3 and PMIP4 models overlaps for all the variables of interest in this study, therefore we think that it is reasonable to treat them as a single ensemble. This same approach is used in numerous studies, e.g. Brown et al. (2020); Wang et al. (2023). The alternative option would be to examine PMIP3 and PMIP4 LGM simulations separately, which we do not think would provide greater insight and which would result in two very small ensembles.

Table 2 does not feel very necessary. The authors could consider moving it to SI to save some space.

[Response]: Table 2 summarises information from Kageyama et al. (2017) for PMIP4 models and from the PMIP3 website for PMIP3 models and therefore we feel that it is worthwhile to include.

L168: Why use both ts and tas?

[Response]: As suggested by reviewer 1, we will change all of our ts analysis in Section 3.1 to surface air temperature (tas) to allow a clearer comparison between our study and previous work.

Fig 4 / Table 3: What is the GMST simulated by these models? Are the models which are cold on a global scale also the ones simulating cold temperatures over Australia?

[Response]: This is a good question. We will add the GMST values to new Figure 3 (reproduced from Table 3) to allow comparison between global temperatures and Australian temperatures and comment on this relationship in the text.

Table 3: Tables are not great to visualize data (also true for Table 4 and 5). The authors could consider a different type of plot to show the reader the large intermodel differences in the seasonal amplitude (not commented in the text?) in a single glance.

[Response]: Thanks for the suggestion. We will change Table 3 to a scatter plot (new Figure 3) which is similar to Figure 1 (b) from Kageyama et al. (2021) to compare the seasonal and global changes in individual models. Table 4 has been removed. We will keep 5 because that it is easier to compare each column in the table to identify differences between precipitation changes and P-E changes and to see values averaged over land only.

L250/L269: Please consider using transitions between subsections (here as in other instances). It is the opportunity to remind the reader of your scientific reasoning (e.g. how these variables are linked).

[Response]: Thanks. We will add transitions as suggested.

L272: I am wondering whether Fig S1 which shows very large model disagreements should not be part of the main text. Also, please explain why you chose to plot the JJA season specifically.

[Response]: The JJA season is when the westerlies bring rainfall to southern Australia. We therefore wanted to see whether there was a shift in the westerlies in this season in order to help understand changes to Australian climate.

We will add Fig S1 to the main text in Section 3.2, and the whole section will be rewritten.

L304-305: 'In JJA the SH westerlies shift equatorward'. Would it be worth it to also investigate the seasonal shift of the westerlies in Sect. 3.2?

[Response]: We explore the shift in the position and intensity of the westerlies in Section 3.2 with a focus on the SH winter (JJA) season. We will add a new figure which shows the MMM change in 850 hPa winds and convergence for each season.

L397-398: Does this correspondance hold for all models?

[Response]: There is higher model agreement over northern Australian precipitation change for MAM than for other seasons – this will be indicated with stippling in the relevant figures. The drying occurs over the region of cooler land, particularly the exposed Sahul shelf in a large majority of models. It is not possible to include seasonal plots for all models for all variables or the paper and Supplement would be unreasonably long.

L412-413: Does this relationship hold if you use the change in strength of westerly winds over the same region?

[Response]: Section 3.3.2.1 (winds) will be rewritten.

L463-465: I will make a subjective comment here. While this is a valid reason to criticize the proxy records (well-explained in introduction), I feel like modellers should maybe not be too critical of proxy uncertainties when such large intermodel differences are observed. The primary reason why we are observing this model-data disagreement might be that, well, models are wrong. I will also point out here that the discussion and especially the conclusion seem lenient with models. I would expect the large intermodel difference observed to reflect a poorly-represented process.

[Response]: Thanks for this important comment. As the reviewer points out, given the model disagreement, it is indeed reasonable to argue that at least some models are wrong. We are interested in finding areas of model-proxy agreement, and areas where models show robust inter-model agreement. However, we will also expand the discussion of possible model biases and uncertainties.

L467-470: Could you discuss the potential reasons why you can find a displacement of the boundary line but no consistent latitudinal shift in westerly winds? Do you have any idea?

[Response]: Section 3.2 and Section 3.3.2.1 (winds) will be rewritten. As noted earlier, the table of boundary lines will be removed, instead, a new figure showing the MMM LGM

seasonal 850 hPa wind change and Fig S1 showing individual model 850 hPa wind change in JJA will be added into Section 3.2.

Fig. 11: Consider different marker styles or colors for individual models or model generation (CMIP5/6).

[Response]: Section 3.3.2.1 (winds) will be rewritten and Fig 11 will be removed.

Fig. 6: It would make sense to put MPI-ESM-P (CMIP5) and MPI-ESM-LR (CMIP6) in the same column on Fig 6 so that it is easier to compare the two generations visually.

[Response]: As discussed earlier, we do not assume that the two generations of a particular climate model will produce a similar simulation of LGM climate, and we do not wish to focus on comparing generations of specific models. Therefore, it is not necessary to put them in the same column on Figure 6.

**Technical corrections**

L16-17: 'many regions' is unclear

[Response]: This sentence will be rewritten to provide more detail of vegetation changes based on proxies and biome models, with citation of Prentice et al. (2011) as the appropriate source: "There was a large reduction in area covered by boreal and temperate forests in northern mid- to high latitudes, expanded lowland tundra in Eurasia, expansion of savanna and grasslands at the margin of Amazon tropical forests and replacement of some areas of tropical forest in Africa, China and Southeast Asia with savanna, woodland and grassland (Prentice et al. 2011)."

Reference:

Prentice, I. C., Harrison, S. P., & Bartlein, P. J. (2011). Global vegetation and terrestrial carbon cycle changes after the last ice age. New Phytologist, 189(4), 988-998. https://doi.org/10.1111/j.1469-8137.2010.03620.x.

L26: 'glaciers' instead of 'glaciation'

[Response]: Thanks. This will be corrected.

L43: 'reflect' or 'are associated with'?

[Response]: We will change the wording as suggested.

L44: Consider replacing 'related to combinations' with something like 'caused by a combination'

[Response]: We will change the wording as suggested.

L45: 190 ppm in Table 2

[Response]: Thanks for pointing it out. The 180 ppm in L45 are summarised from proxy records, whereas Table 2 shows the IPCC configuration values for the LGM experiment.

L110: 'drier conditions' would work but not 'drier changes'

[Response]: This will be modified.

L121: 'differences' or 'gaps' would fit better than 'variations'

[Response]: Thanks, this will be modified.

L124: 'more recent' instead of 'newer'

[Response]: We will change the wording as suggested.

L144-145: 'Furthermore, the PMIP4 protocol highlighted…'

[Response]: (Line 155) We will change the wording as suggested.

L170: 10 m

[Response]: (Line 172) We will change the wording as suggested.

L328: What is the Top-End region?

[Response]: We will change this to "central northern Australia" as the region "Top End" is only known in Australia.

---

## Author Comment (AC3)

**Reviewer#3:**

**General comments**

Climate change in the Southern Hemisphere is poorly understood, and large model biases are known to exist. Studying how climate has changed at the LGM may provide unique insights into the climate dynamics of this region. This manuscript investigates changes in temperature, precipitation, and wind over Australia at the LGM using a subset of PMIP3 and PMIP4 models and compares these changes to existing proxy data. Such a study could be helpful in improving our understanding of Australian climate.

The analysis is generally okay: the authors looked at the climate response in individual models, ensemble mean, and seasonality. However, I think the authors could have added some more in-depth analysis or discussion. One thing they can do is to expand the inter-model agreement (hatching the maps of ensemble mean could be helpful), and consider how model disagreement may affect the ensemble mean values.

[Response]: Thank you for your comments and suggestions. Stippling will be added to all MMM figures (Figure 3, 5, 8, 9, 10) to show model agreement. All other figures show individual models which allows model consistency to be evaluated by the reader.

I also think that the mechanisms for changes in temperature, precipitation, and wind are not adequately discussed. Please see my specific comments.

In addition, I think the authors should do their due diligence to acquire model output from all PMIP4 models.

In terms of presentation, the manuscript is structured logically. But the color scales for showing hydroclimatic anomalies could be improved such that the map colors are not overwhelmed by the changes at the coast to make it easier to see changes over the continent. And a better integration of data-model comparison could be achieved by showing the proxy-reconstructed changes in the map of simulated changes.

[Response]: Thank you for the suggestion. We have tried to modify the color scales of hydroclimate plots by changing to a smaller range of colorbar so that the values over land are more easily seen. However, it is difficult choose a color scale for hydroclimate figures which allows all areas to be clearly seen. We include Table 5 to show average changes over land for this reason.

Regarding the proxy-model comparison, as noted for reviewer 1, we have provided already that the sign of the change is uncertain, especially when taking into account the $CO_2$ effect on vegetation records, and therefore we refer to the literature but don't include any proxy records in our plots.

**Specific comments**

The Abstract ends abruptly by describing changes in winds, whereas here it should provide the readers with some key implications or take-home message of this paper.

[Response]: A sentence will be added to the Abstract summarising key results.

47: Ujvari et al 2018 is not an appropriate reference, as it does not talk about changes in dust at the LGM.

[Response]: Thanks. This reference will be removed.

61: Many of these referenced papers did not use PMIP4.

[Response]: This sentence will be corrected to refer to PMIP4 studies only.

66: You did not mark these regions discussed here in Figure 2. Maybe use consistent terminology here as the rest of the paper.

[Response]: Thanks for the suggestion. We will change the naming in Section 1.1 to the consistent name as the rest of the paper.

74: Reference for the fire study?

[Response]: The information comes from Rowe et al. (2020) who examined microcharcoal in the Girraween lake sediment record as an indicator of landscape fire. This sentence follows the previous sentence summarising results from Rowe et al. (2020) but we will add a second citation of the paper in this sentence.

77: You cited a wrong Denniston et al (2013) paper. The correct one is:

Denniston, R. F., Wyrwoll, K. H., Asmerom, Y., Polyak, V. J., Humphreys, W. F., Cugley, J., ... & Greaves, E. (2013). North Atlantic forcing of millennial-scale Indo-Australian monsoon dynamics during the Last Glacial period. Quaternary Science Reviews, 72, 159-168.

Note that in the paper you cited, the C126 speleothem shows more positive d18O and d13C values at LGM than the late Holocene, which might suggest drier glacial conditions.

[Response]: Apologies. The correct reference will now be provided, and the sentence modified to better reflect the information shown in the speleothem.

143: This statement is incorrect: Zhu et al. (2021) only assessed CESM2-CAM6, the "low top" version of CESM2, not the WACCM version.

[Response]: We apologise for the incorrect statement. We had some trouble linking the available CESM2 model simulations on ESGF with documentation and relevant publications. We will now include the CESM2-WACCM model as we now understand this model does not have an unrealistic climate sensitivity.

156: Do these different ice sheet configurations affect the Australian climate at LGM? Did you use them in your study?

[Response]: A new Table will be added which gives information of ice-sheet reconstructions for individual models. The PMIP3 models used PMIP3 ice-sheet configurations and the four PMIP4 models used in this study all used the "ICE-6G_C" ice-sheet reconstruction. There will be influences on the simulated LGM climate affected by the different ice sheet configurations between two model generations.

180: Why do you choose the first 100 years? Models need time to reach new climate equilibrations in response to external forcings. I would use the last 100 years if possible at all.

[Response]: Thanks for the suggestion. This has been justified for reviewer 1 and 2 as well. We are using the first 100 years due to the reason that the simulations public on ESGF are already in equilibrium so there will be no significant differences for whether it is the first or the last 100 years. In many cases, only 100 years were available from ESGF.

185: specify it is austral summer/winter. I also think this is where you can describe the regional climate systems in more detail. i.e., winter precipitation in the south is associated with the westerlies, summer precipitation in the north is associated with the monsoon.

[Response]: Thanks for the suggestion. We will expand the description of the regional climate systems.

241: If "land areas warm more than surrounding oceans" during DJF and SON is the case, why DJF and SON show opposite signs in temperature change over Sahul? Are there other mechanisms that could cause this change in temperature?

[Response]: The two paragraphs discussing Figure 5 will be rewritten to clarify the results. There are a number of points which required better explanation.

245-250: How do these analyses relate to your results in Figure 5? If there is enhanced cooling in SON and reduced cooling in MAM, why Fig 5 shows more cooling in MAM and less cooling in SON?

[Response]: The two paragraphs discussing Figure 5 will be rewritten to clarify the results. There are a number of points which required better explanation.

311: What is this "SST gradient"?

[Response]: The discussion refers to surface temperature gradients in the region. It will be rewritten to clarify.

395-396: This statement does not make sense. Fig 5 shows DJF cooling and SON warming over northern Australia, why does it case wetting in both seasons? What is the "response to changes in seasonal heating" and "changes in atmospheric circulation" here?

[Response]: The discussion of drivers of change in rainfall will be rewritten to clarify. New figures showing changes in 850 hPa winds in Section 3.2 will assist to show the relevant processes – due to changes in offshore/onshore circulation.

414: p = 0.082 suggests that the correlation is not significant or "moderate" – it is insignificant. By the way, I wonder how do changes in precipitation and the northward displacement of easterly-westerly boundary correlate.

According to your findings, what is the mechanism for changes in winds?

[Response]: Thank you, we agree this is insignificant. The whole Section 3.3.2.1 (winds) will be rewritten. The mechanism for changes in winds will be discussed with reference to other LGM westerly studies.

**Technical corrections**

268: You don't need a 3.2.1 subsection here

[Response]: This will be corrected.

323: Figure S4 is MMM seasonal anomalies for LGM - PI evapotranspiration, not precipitation.

[Response]: Thanks. This will be corrected.

397: to the => to the

[Response]: This will be corrected.

403: should be 3.3.2.2

[Response]: This will be corrected.

---

## Author Response (AR1)

**General response:** We thank the reviewer for their detailed and thoughtful comments. We have revised the text in response to the review comments, as outlined below.

The knowledge of the characteristics and mechanism for the climate change over Australian regions in LGM is still not enough. This study investigated the climate changes at the LGM over the Australian region, in terms of temperature, precipitation, moisture balance and wind, based on the output from PMIP3 and PMIP4 simulations. The work might contribute to our understanding of the hydrological change of Australia in ice ages. The following are my comments and reviews for the authors' consideration.

*Comments:*

1. The uncertainties of model simulations and reconstructions are important information for model-data comparison. It would be better to evaluate the model consistence since there was large model spread. Specifically, the authors could further provide the percentage of model ensembles consistent on of the signal of their multiple model ensembles mean value.

**Response:** Stippling has been added to all MMM temperature and precipitation anomaly figures (Figure 2, 4, 10, 11, 12) to show model agreement. All other figures show individual models which allows model consistency to be evaluated by the reader.

For the reconstruction, the background information of proxy used here and their uncertainty could be listed in a table. The information of the LGM climate getting wetter or drier and in which parts of Australia based on proxies is still not clear, even though the authors cited others' work in line 459-463. It would be easier to read and make comparison if were there reconstructed data mapped on the plots of model results.

**Response:** We emphasize that this study is not concerned specifically with detailed data-model comparison of LGM hydroclimate in Australia. Our principal reason for avoiding a quantitative data-model comparison with proxy climate reconstructions of the Australian LGM is that the Australian proxy-based LGM palaeoclimate literature typically has provided qualitative reconstructions, ('drier', 'much drier', 'somewhat drier') and that, where quantitative reconstructions have been provided, they have uniformly been based on comparison of LGM plant distributions with modern plant distributions, or comparison of the LGM occurrence of mobile sand dunes with their modern distribution, respectively. These comparisons, however, have ignored the plant physiological effects of low atmospheric $CO_2$, which is increasingly recognised as a problem that should not be ignored (Scheff et al., 2017; Prentice et al., 2022). In brief, C3 plants perceive an LGM world with low (180 ppm) $CO_2$, as much 'drier' than today, so that model simulations with dynamic vegetation typically show widespread forest reduction, even when holding temperature and precipitation at modern values. Moreover, it is increasingly suspected (e.g. Scheff et al., 2017; Roderick et al., 2015) that greater LGM dustiness and dune-mobilisation are secondary effects from reduced plant productivity, via landscape destabilisation.

For these reasons, we believe it is misleading to describe or list the published Australian LGM hydroclimatic reconstructions in detail. We believe it is clearer, to explain in summary

form the recently-recognised problems associated with understanding LGM hydroclimate, as described above; and treat the published reconstructions collectively, at the level of a literature, rather than as individual reconstructions.

1. Usually, modeling community use surface air temperature (SAT) instead of surface temperature (ts) to investigate the temperature change, and to explain the related change of circulation and/or precipitation.

**Response:** We have redone all temperature analysis for surface air temperature (tas) instead of surface temperature (ts).

2. The climate proxy from 28 to 18 ka is compared with the LGM simulations at 21ka (Line 55-56). This may also contribute to the model-data inconsistence considering the extended date of proxy. For example, the variability of climate proxy during 28 and 18 ka may switch between drier or wetter condition than pre-industry and thus make the complexity of model-data comparison. This point could discussed further when necessary

**Response**: We have checked the dates of the proxy records, this range can be narrowed to 24-18 ka, which is consistent with the definition of the LGM in many previous studies (e.g. Clark et al., 2009).

3. As pointed out by the authors, the sst gradients and related circulation change could explain the precipitation change (Line 309-311). Thus the analysis of sst change (which roughly equals to the ts value over ocean) and model-data comparison of sst could improve the knowledge of the LGM climate change over Australia.

**Response**: We have added Supplementary Figure S8 showing the surface temperature change for each model over the northern part of the domain. We include both land and ocean surface temperature to show any changes in the land-ocean temperature gradient. The link between these surface temperature changes, precipitation and circulation changes is now discussed in Section 3.3.2.

***Line by line comments***

Line 42. "Many regions", could be pointed in details.

**Response**: This sentence was rewritten to provide more detail of vegetation changes based on proxies and biome models, with citation of Prentice et al. (2011) as the appropriate source. It has been changed to: "There was a large reduction in area covered by boreal and temperate forests in northern mid- to high latitudes, expanded lowland tundra in Eurasia, expansion of savanna and grasslands at the margin of Amazon tropical forests and replacement of some areas of tropical forest in Africa, China and Southeast Asia with savanna, woodland and grassland (Prentice et al. 2011)."

Reference:

Prentice, I. C., Harrison, S. P., & Bartlein, P. J. (2011). Global vegetation and terrestrial carbon cycle changes after the last ice age. *New Phytologist*, 189(4), 988-998. https://doi.org/10.1111/j.1469-8137.2010.03620.x.

Line 92-100. The reconstructed evidence of moisture or hydrocliamte could be compared with model simulations in the section of discussions.

**Response**: The comparison with hydroclimate proxies included in Section 4.2 is now expanded.

Line 155-157. There were three different ice sheet reconstructions. Thus it's necessary to clarify the information of ice sheet configuration of the four models from PMIP4.

**Response:** An expanded version of Table 1 has been updated adding one more column listing the ice-sheet reconstructions in individual models. The four PMIP4 models analysed in this study used the ICE-6G_C ice sheet configuration and the eight PMIP3 models used PMIP3 ice-sheet reconstruction (differences between the two LGM ice-sheet reconstructions in Kageyama et al., 2017). This is now mentioned in Section 2.1.

Table 2. In term of vegetation of PMIP4, were there any model using the dynamic vegetation? Please check and make it clear.

**Response:** Only the CMIP6 AWI-ESM-1-1-LR model is using dynamic vegetation. This is now noted in the text in Section 2.1.

Line 180-182. Usually models use the last 100 years, instead of the first 100 years, to do analysis. Were there any big differences between those two choices?

**Response:** We apologise for the confusion with the wording of this sentence. We now clarify why we chose the first 100 years of the model run in Section 2.2. We also provide a brief explanation below.

Most climate models used in this study only have 100-year length of simulation based on the number of years of data available on the ESGF (Earth System Grid Federation). According to Kageyama et al. (2017), the models have all been spun-up until equilibrium following the PMIP protocols (refer to Kageyama et al. (2017) for details of the spin-up requirements). At least 100-year data from the equilibrium part of the simulation is required to store on the ESGF (Kageyama et al., 2017). Therefore, the data stored on the ESGF has already reached equilibrium and it is reasonable to analyse the first 100 years (there should be no significant drift, as per PMIP4 protocols).

Line 213-214. The difference between the analyses in the paper with Kageyama et al. may lies in the choice of ts, instead of SAT. Please check.

**Response:** Thank you for the suggestion. As you note, Kageyama et al. (2021) used surface air temperature instead of ts. We have changed all of our temperature analysis to surface air temperature (tas) to allow a clearer comparison between our study and previous work.

**Response to Reviewer 2:**

**General response:** We thank the reviewer for their comprehensive and constructive comments on the manuscript. We have revised the text in response to the review comments, as outlined below.

**General comments**

In this study by Du et al., the authors propose a modelling intercomparison study of the simulated surface temperature, precipitation, moisture balance and winds at the LGM, focusing on the Australian region.

Such a study would fit well within the scope of Climate of the Past. The case study of Australia is interesting for several reasons such as the impact of sea level change on coastlines, the specific location (SH, in proximity to the Maritime Continent and Southern Ocean), and the overall disagreement of models and proxies especially concerning the latitudinal shift and variation of strength of the westerly winds, which seem both poorly represented in PMIP models (Chavaillaz et al., 2013) and poorly constrained by conflicting paleodata records (Kohfeld et al., 2013). The manuscript is also clear and an easy read.

However, as such (and as is often the case with intercomparison studies), the paper reads as very descriptive and superficial, so we are struggling to learn something new. Knowledge gaps, uncertainties in both model and data and processes/mechanisms are either barely mentioned or simply not highlighted enough so when we reach the end, the impression is underwhelming.

I am providing more concrete illustrations as to what could be improved and how with points 1-3. I also have suggestions related to methodology in points 4-8. So I would like to recommend a number of improvements before publication, in the hope of helping give this study more weight.

1. Please elaborate on the reasons why a case study of the past climatic changes over Australia is interesting. It would be great to mention climate processes that are key in this region. An example since dust transport is mentioned (L93): does this aridity have the potential to significantly enhance iron fertilisation in the ocean?

**Response:** In the Introduction (Section 1), we have now added more discussion of why this case study is interesting with reference to key climate processes in the region. While the issue of dust transport and iron fertilisation is beyond the scope of the present study, we now include discussion of the major changes in Southern Hemisphere climate and circulation which may have influenced Australia during the LGM and also the relevance to interpretation of records of human occupation and changes in biogeography at this time.

2. Consider adding one or two last sentences to the abstract and a paragraph in the discussion/conclusion to give the reader a broader perspective and hindsight on what we have learned and how significant are these new findings. A few questions to help brainstorm: So what? In the basis of the existing literature and these new findings, have we achieved a better understanding of the processes which influenced the past Australian regional climate? If not, what are we missing? Do we understand the model response to LGM forcings? What does it entail?

**Response:** Thank you for your useful suggestions. We have substantially edited the Abstract and Discussion to include more discussion of the significance of the findings.

3. Please underline the knowledge gaps in the introduction. As such, the introduction is very descriptive (not impactful), with a structure (global changes / changes in different regions of Australia) that doesn't help guide the reader very logically towards understanding the knowledge gaps, their importance, the scientific question and the methodology used in this study. If the authors would like to keep this regional description structure, then it would be welcome to also point out the contrasts between these different regions, and also with the global climate, with a few short sentences to conclude this subsection.

**Response**: The Introduction has been substantially rewritten to more clearly explain the significance of the study in relation to existing knowledge gaps. The proxy section (Section 1.1) retains the structure of northern/southern/central Australia as this follows many previous proxy-based studies examining regional changes in Australian climate during the LGM (e.g. Reeves et al., 2013; Fitzsimmons et al., 2013; Petherick et al., 2013, etc). We have now added a summary paragraph to Section 1.1 explaining the agreement and disagreement between regions and records.

Still, I feel like the sometimes conflicting proxy records and the 'uncertainty about the drivers of the LGM climate changes' (L74-75), the 'ambiguous results' of models (L117) should be arguments brought to the reader's attention in a more convincing order to justify the need of this particular intercomparison study and its methods. For exemple, it is not clear what is the advantage of using an intercomparison method, nor for which reasons modellers simulate the LGM period (L102-107). It is also not clear why the authors chose to examine these three specific climate variables. I believe there are ways to reinforce the visibility of the scientific reasoning behind this approach.

**Response**: We note that a model intercomparison approach is widely used in numerous studies of PMIP LGM and other PMIP simulations. We examine the most common variables required to characterise climate (temperature, precipitation, wind). We now more clearly explain the specific motivation for the study in Section 1 and the design of the study in Section 2.

4. Only PMIP4 outputs available on the ESGF were included. This is a bit of a shame because it limits comparison with the Kageyama et al. (2021) results, and the model ensemble size (and therefore the robustness of the results). CESM2 is also excluded for very good reasons (L143-144), but I believe the authors have found the source of the exaggerated cooling in a cloud microphysics parameterization and run a corrected simulation. This is of course up to the authors, but they could consider contacting both the Kageyama et al. (2021) and Zhu et al. (2021) authors to request the model outputs. This would also enable the authors to compare individual model versions (CMIP5 vs CMIP6) and discuss potential improvements between the two generations. For now, only MPI-ESM is in the two ensembles.

**Response:** Thanks for your comments. We decided to make use of only those models which are publicly available via the ESGF to ensure our results were readily reproducible. We follow the standard approach in CMIP climate model studies which typically only use publicly archived simulations. Many other PMIP-based studies have also used this approach, e.g. a recent paper on LGM ITCZ changes from PMIP3/4 models used a similar set of models (Wang et al., 2023).

Regarding CESM2 models, we had some trouble linking the available CESM2 model simulations on ESGF with documentation and relevant publications. We now include the CESM2-WACCM-FV2 model as this model has pr, tas and ua, va variables available on ESGF

(CESM2- FV2 did not have the "pr" variable). All figures and tables have been updated to include this extra model.

The large-scale comparison between CMIP5 and CMIP6 models was provided in Kageyama et al. (2021) so we do not need to repeat this analysis. We do not assume that the two generations of a particular climate model (e.g. MPI-ESM) will produce a similar simulation of LGM climate, and we do not wish to focus on comparing generations of specific models – this is typically done by the relevant modelling groups.

The focus of the present study is to summarise the simulation of LGM climate of Australia from available PMIP3 and PMIP4 models. This information is of interest to the Australian palaeoclimate community and will provide a basis for comparison with numerous proxy-based reconstructions of standard climate variables such as surface temperature and precipitation, as outlined in the Introduction.

Reference:

Wang, T., Wang, N., & Jiang, D. (2023). Last glacial maximum ITCZ changes from PMIP3/4 simulations. Journal of Geophysical Research: Atmospheres, 128, e2022JD038103. https://doi. org/10.1029/2022JD038103

5. The authors mentions that all model outputs are regridded (L170), and it seems that all the following analysis use these regridded outputs. Of course, this is necessary to compute and plot the multimodel mean, but I am wondering whether it would be worth extending the use of the model native grids when plotting the individual maps and wind profiles. Is the latitudinal shift of westerly winds affected by the model resolution? Are there differences in the land-sea mask of each model which could impact the simulated temperature and precipitation patterns? Please consider plotting the PI or LGM land-sea mask (e.g. as a grey or dashed contour) on maps.

**Response:** Thanks for the suggestion. We have recalculated boundary lines between SH easterlies and westerlies on the native model grids (old Table 4), however, we decided to remove it from the paper. This is because the results differ quite a lot on the different grids, and we think they are not robust enough to be included. The whole Section 3.2 has been rewritten, including adding Figure 6 showing MMM seasonal wind changes and Figure 7 showing JJA wind changes for each model. This allows a more general discussion of changes in winds without relying on metrics of the position of the westerlies.

Regarding the influences from the land-sea masks in different models, we now plot the LGM land masks for individual models in Supplementary Figure S1 to show the different LGM land areas configured in each model. We decided not to plot the land masks over the maps as this would make it difficult to see the details of the variables plotted.

6. On maps, the authors should also represent the multimodel agreement significance as hashes (when >90%) and the proxy data as scatter points, whenever quantitative reconstructions can be found. I consider important that the reader is able to compare visually the performance of the models with the available proxy data, especially as the authors conclude that there is a general good agreement between model and data.

**Response:** We have now added stippling to all MMM figures to show areas of model agreement. As noted in our response to reviewer 1, we believe that the sign of the change from many hydroclimate proxy records is uncertain, especially when taking into account the

CO$_2$ effect on vegetation records, and therefore we refer to the literature but don't include proxy records in our plots.

7. The authors mention using the first 100 years of simulation for the analysis. Why not the last 100 years? Please check that the simulations are in equilibrium, e.g. by computing the drifts.

**Response:** As for Reviewer 1, most climate models used in this study only have 100-year length of simulation based on the number of years of data available on the ESGF (Earth System Grid Federation). According to Kageyama et al. (2017), the models have been spun up until equilibrium following the PMIP protocols (refer to Kageyama et al. (2017) for details of the spin-up). At least 100-year data from the equilibrium part of the simulation is required to store on the ESGF (Kageyama et al., 2017). Therefore, the data stored on the ESGF is already in equilibrium. We do not believe it is necessary to calculate model drifts ourselves as this has presumably been done by the PMIP modelling groups.

8. The paper would deserve more quantifications. An example is in L273: 'Some models show weakening and other model show strengthening but there are other instances (e.g. L372). It would be great to provide precise figures, e.g. phrasings like '5 models out of 12 show a weakening of at least 20%...'

**Response:** To provide a quantitative comparison as suggested, we would need to calculate the change in winds over a particular domain. The spatial pattern of wind changes as shown in new Figure 7 is quite heterogenous, with some models showing regions of both increased and decreased zonal winds over southern Australia and the adjacent ocean. Given this complex spatial response, we do not want to provide a subjective "area average" wind change, but simply to point out the lack of model agreement for changes in zonal wind strength over this domain. Therefore, we think the qualitative description is adequate.

**Specific comments**

L2, L12 and L13: 'changes at the LGM', 'to cool by 2.6 at the LGM' and ,'decreased'. Unlike in the rest of the paper, it is sometimes unclear in the abstract that we are mentioning changes relative to the PI. This has to be indicated in some way (e.g. LGM-PI anomaly, with respect to PI...) or else the verbs are inconsistent with the time direction.

**Response:** The wording has been changed to avoid confusion.

L13-14: Why are the changes in temperature and precipitation indicated over two different defined regions?

**Response**: The calculation for MMM precipitation changes was over a smaller domain aiming to exclude the land areas between 0 to 10°S to allow comparison with previous studies focused on northern Australian rainfall. The has been removed from the Abstract to make way for other material.

L18-19: I find the sentence explaining the potential reasons for model-data disagreement to be rather vague and could be reformulated.

**Response:** The Abstract, including this sentence, has been rewritten to improve clarity as suggested.

L23: The time window proposed for the LGM is unusual and only justified later in the text.

**Response:** We have changed the LGM time to "~21,000 years ago" here to avoid confusion and later introduce the extended LGM concept.

L31: Why such a gap between the Annan et al. (2022) and the Tierney et al. (2020a) estimates?

**Response:** This disagreement is due to the "choice of prior", i.e. the particular climate model used in the Tierney et al. (2020) study, as discussed by Annan et al. (2022). We now briefly note the dependence on method.

Fig 1 and 2: These figures do not bring a lot of information to the table. The authors could consider combining them, combining Fig 1 with e.g. Fig 3 (the land-sea mask could be indicated on another map with a grey contour), or enriching them with more information (e.g. the Sunda and Sahul shelves mentioned in L34 could be annotated on the map to show the reader their location). As for Fig 2, consider using a different style (than the red contour, e.g. hashes) for the southwest box.

**Response:** Thanks for the suggestion. We have combined Figure 1 and Figure 2 to make a new Figure 1 with the LGM land mask for CCSM4 model as background shading. The southwest box is plotted in hashes.

L85: The authors could elaborate on the reasons why they used the word 'possibly'.

**Response:** According to Kohfeld et al. (2013), "If a single cause related to the southern westerlies is sought for all the evidence presented, then an equatorward displacement or strengthening of the winds would be consistent with the largest proportion of the observations." This is now rewritten as "perhaps consistent with" to summarise the findings of the Kohfeld et al. (2013) study.

L110: 'not fundamentally different' could be true for the variables examined in the study and not others. Please check if this is the case for all variables (including ocean circulation).

**Response:** This paragraph is now deleted. However, in relation to combining the PMIP3/CMIP5 and PMIP4/CMIP6 models into a single ensemble, we feel that this is a reasonable approach given that the sample spread of PMIP3 and PMIP4 models overlaps for all the variables of interest in this study (except for CESM2-WACCM-FV2 temperature). This same approach is used in numerous studies, e.g. Brown et al. (2020); Wang et al. (2023). The alternative option would be to examine PMIP3 and PMIP4 LGM simulations separately, which we do not think would provide greater insight and which would result in two very small ensembles.

Table 2 does not feel very necessary. The authors could consider moving it to SI to save some space.

**Response:** Table 2 summarises model boundary condition information from Kageyama et al. (2017) for PMIP4 models and from the PMIP3 website for PMIP3 models and therefore we feel that it is worthwhile to include.

L168: Why use both ts and tas?

**Response:** We have now changed all temperature analysis to "tas" rather than "ts" to allow a clearer comparison between this study and previous work (following Reviewer 1 recommendation).

Fig 4 / Table 3: What is the GMST simulated by these models? Are the models which are cold on a global scale also the ones simulating cold temperatures over Australia?

**Response:** This is a good question. We have added the GMST values to new Figure 5 (reproduced from Table 3) to allow comparison between global temperatures and Australian temperatures and also comment on this relationship in the text in Section 3.1.

Table 3: Tables are not great to visualize data (also true for Table 4 and 5). The authors could consider a different type of plot to show the reader the large intermodel differences in the seasonal amplitude (not commented in the text?) in a single glance.

**Response:** Thanks for the suggestion. We have changed Table 3 to a scatter plot (new Figure 5) which is similar to Figure 1b from Kageyama et al. (2021) to compare the seasonal and global changes in individual models. Table 4 has been removed. We retain Table 5 (new Table 3) because that it is easier to compare each column in the table to identify differences between precipitation changes and P-E changes and to see values averaged over land only.

L250/L269: Please consider using transitions between subsections (here as in other instances). It is the opportunity to remind the reader of your scientific reasoning (e.g. how these variables are linked).

**Response**: Thank you for the suggestion. We have tried to add transitions between subsections where appropriate.

L272: I am wondering whether Fig S1 which shows very large model disagreements should not be part of the main text. Also, please explain why you chose to plot the JJA season specifically.

**Response:** The JJA season is when the westerlies bring rainfall to southern Australia, as explained in the text. We therefore wanted to see whether there was a shift in the westerlies in this season in order to help understand changes to Australian climate.

Old Figure S1 has been added to the main text (new Figure 7 in Section 3.2), and the whole section has been rewritten.

L304-305: 'In JJA the SH westerlies shift equatorward'. Would it be worth it to also investigate the seasonal shift of the westerlies in Sect. 3.2?

**Response:** We explore the shift in the position and intensity of the westerlies in Section 3.2 with a focus on the SH winter (JJA) season. We have also added a new figure (new Figure 6) which shows the MMM change in 850 hPa winds and convergence for each season.

L397-398: Does this correspondance hold for all models?

**Response:** There is higher model agreement over northern Australian precipitation change for MAM than for other seasons – this is now indicated with stippling in the relevant figures. The drying occurs over the region of cooler land, particularly the exposed Sahul shelf in a large majority of models. It is not possible to include seasonal plots for all models for all variables or the paper and Supplement would be unreasonably long.

L412-413: Does this relationship hold if you use the change in strength of westerly winds over the same region?

**Response:** This Section has been rewritten and the scatter plot removed as the relationship was not statistically significant.

L463-465: I will make a subjective comment here. While this is a valid reason to criticize the proxy records (well-explained in introduction), I feel like modellers should maybe not be too critical of proxy uncertainties when such large intermodel differences are observed. The primary reason why we are observing this model-data disagreement might be that, well, models are wrong. I will also point out here that the discussion and especially the conclusion seem lenient with models. I would expect the large intermodel difference observed to reflect a poorly-represented process.

**Response:** Thanks for this important comment. As you point out, given the model disagreement, it is indeed reasonable to argue that at least some models are wrong. We are interested in finding areas of model-proxy agreement, and thus focus on areas where models show robust agreement. However, we now also expand the discussion of possible model biases and uncertainties.

L467-470: Could you discuss the potential reasons why you can find a displacement of the boundary line but no consistent latitudinal shift in westerly winds? Do you have any idea?

**Response**: Section 3.2 and Section 3.3.2.1 (winds) has been rewritten. As noted earlier, the table of boundary lines was removed (old Table 4), instead, a new figure showing the MMM LGM seasonal 850 hPa wind change (new Figure 6) and the old Figure S1 showing individual model 850 hPa wind change in JJA (new Figure 7) were added into Section 3.2.

Fig. 11: Consider different marker styles or colors for individual models or model generation (CMIP5/6).

**Response**: Section 3.3.2.1 (winds) has been rewritten and the old Figure 11 has been removed from the text to the new Supplementary Figure S9.

Fig. 6: It would make sense to put MPI-ESM-P (CMIP5) and MPI-ESM-LR (CMIP6) in the same column on Fig 6 so that it is easier to compare the two generations visually.

**Response**: As discussed earlier, we do not assume that the two generations of a particular climate model will produce a similar simulation of LGM climate, and we do not focus on comparing generations of specific models in this study. Therefore, we prefer to arrange models alphabetically within CMIP generations.

**Technical corrections**

L16-17: 'many regions' is unclear

**Response:** This sentence was rewritten to provide more detail of vegetation changes based on proxies and biome models, with citation of Prentice et al. (2011) as the appropriate source. It has been changed to: "There was a large reduction in area covered by boreal and temperate forests in northern mid- to high latitudes, expanded lowland tundra in Eurasia, expansion of savanna and grasslands at the margin of Amazon tropical forests and replacement of some areas of tropical forest in Africa, China and Southeast Asia with savanna, woodland and grassland (Prentice et al. 2011)."

Reference:

Prentice, I. C., Harrison, S. P., & Bartlein, P. J. (2011). Global vegetation and terrestrial carbon cycle changes after the last ice age. *New Phytologist*, 189(4), 988-998. https://doi.org/10.1111/j.1469-8137.2010.03620.x.

L26: 'glaciers' instead of 'glaciation'

**Response:** Thanks. This was corrected.

L43: 'reflect' or 'are associated with'?

**Response:** We have changed the wording as suggested.

L44: Consider replacing 'related to combinations' with something like 'caused by a combination'

**Response**: We have changed the wording as suggested.

L45: 190 ppm in Table 2

[Response]: Thanks for pointing it out. The 180 ppm in L45 are summarised from proxy records, whereas Table 2 shows the IPCC configuration values for the LGM experiment.

L110: 'drier conditions' would work but not 'drier changes'

**Response:** This sentence was deleted.

L121: 'differences' or 'gaps' would fit better than 'variations'

**Response:** We changed the wording to "disagreements".

L124: 'more recent' instead of 'newer'

**Response:** Changed as suggested.

L144-145: 'Furthermore, the PMIP4 protocol highlighted…'

**Response:** (Line 155) Changed as suggested.

L170: 10 m

**Response:** (Line 172) Changed as suggested.

L328: What is the Top-End region?

**Response:** We have changed this to "central northern Australia" as the term "Top End" is only known in Australia.

**General response:** We thank the reviewer for their helpful comments on the manuscript. We have revised the text in response to the review comments, as outlined below.

**General comments**

Climate change in the Southern Hemisphere is poorly understood, and large model biases are known to exist. Studying how climate has changed at the LGM may provide unique insights into the climate dynamics of this region. This manuscript investigates changes in temperature, precipitation, and wind over Australia at the LGM using a subset of PMIP3 and PMIP4 models and compares these changes to existing proxy data. Such a study could be helpful in improving our understanding of Australian climate.

The analysis is generally okay: the authors looked at the climate response in individual models, ensemble mean, and seasonality. However, I think the authors could have added some more in-depth analysis or discussion. One thing they can do is to expand the inter-model agreement (hatching the maps of ensemble mean could be helpful), and consider how model disagreement may affect the ensemble mean values.

**Response:** Thank you for your suggestions. Stippling has been added to all MMM temperature and precipitation anomaly figures (Figure 2, 4, 10, 11, 12) to show model agreement. We also expand the discussion of model disagreement in Sections 3 and 4.

I also think that the mechanisms for changes in temperature, precipitation, and wind are not adequately discussed. Please see my specific comments.

**Response:** Thank you for your comment. We agree that there were inadequate discussions of the changes in temperature, precipitation and winds. We have substantially rewritten these sections, and added new figures showing seasonal changes in winds and wind convergence which expand the discussion of mechanisms. We have responded to your specific comments below.

In addition, I think the authors should do their due diligence to acquire model output from all PMIP4 models.

**Response:** Thanks for your comments. As noted to reviewer 2, we decided to make use of only those models which are publicly available via the ESGF to ensure our results were readily reproducible. We follow the standard approach in CMIP climate model studies which typically only use publicly archived simulations. Many other PMIP-based studies have also used this approach. We have now added CESM-WACCM-FV2 to the CMIP6 set of models and revised the figures and discussion accordingly.

In terms of presentation, the manuscript is structured logically. But the color scales for showing hydroclimatic anomalies could be improved such that the map colors are not overwhelmed by the changes at the coast to make it easier to see changes over the continent. And a better integration of data-model comparison could be achieved by showing the proxy-reconstructed changes in the map of simulated changes.

**Response:** Thank you for the suggestion. We have tried to modify the color scales of hydroclimate plots by changing to a smaller range of colorbar so that the values over land are more easily seen. However, it is difficult choose a color scale for hydroclimate figures which allows all areas to be clearly seen. We include the old Table 5 (new Table 3) to show average changes over land for this reason.

Regarding the proxy-model comparison, as noted for reviewer 1, we have noted already that the sign of the change is uncertain, especially when taking into account the $CO_2$ effect on vegetation records, and therefore we refer to the literature but don't include any proxy records in our plots.

**Specific comments**

The Abstract ends abruptly by describing changes in winds, whereas here it should provide the readers with some key implications or take-home message of this paper.

**Response:** The Abstract has been rewritten to more clearly summarise the key results of the paper.

47: Ujvari et al 2018 is not an appropriate reference, as it does not talk about changes in dust at the LGM.

**Response:** This reference has been removed.

61: Many of these referenced papers did not use PMIP4.

**Response:** This sentence has been corrected to refer to PMIP4 studies only.

66: You did not mark these regions discussed here in Figure 2. Maybe use consistent terminology here as the rest of the paper.

**Response:** Thanks for the suggestion. We have changed the naming in Section 1.1 to follow the same terminology as in the Figure and the rest of the paper.

74: Reference for the fire study?

**Response:** The information comes from Rowe et al. (2020) who examined microcharcoal in the Girraween Lake sediment record as an indicator of landscape fire. This sentence follows the previous sentence summarising results from Rowe et al. (2020) but we now add a second citation of the paper in this sentence.

77: You cited a wrong Denniston et al (2013) paper. The correct one is:

Denniston, R. F., Wyrwoll, K. H., Asmerom, Y., Polyak, V. J., Humphreys, W. F., Cugley, J., ... & Greaves, E. (2013). North Atlantic forcing of millennial-scale Indo-Australian monsoon dynamics during the Last Glacial period. Quaternary Science Reviews, 72, 159-168.

Note that in the paper you cited, the C126 speleothem shows more positive d18O and d13C values at LGM than the late Holocene, which might suggest drier glacial conditions.

**Response:** Apologies. The correct reference is now be provided, and the sentence modified to better reflect the information shown in the speleothem study.

143: This statement is incorrect: Zhu et al. (2021) only assessed CESM2-CAM6, the "low top" version of CESM2, not the WACCM version.

**Response:** We apologise for the incorrect statement. We had some trouble linking the available CESM2 model simulations on ESGF with documentation and relevant publications. We now include the CESM2-WACCM-FV2 model as we now understand this model does not have an unrealistic climate sensitivity.

156: Do these different ice sheet configurations affect the Australian climate at LGM? Did you use them in your study?

**Response:** A new Table has been added which gives information on ice-sheet reconstructions for individual models. The PMIP3 models used PMIP3 ice-sheet configurations and the four PMIP4 models used in this study all used the "ICE-6G_C" ice-sheet reconstruction. There may be influences on the simulated LGM climate due to the different ice sheet reconstructions used, although this is not likely to be large given that Australia is remote from the regions of expanded ice sheets. We therefore do not focus on comparing the role of difference ice sheet reconstructions.

180: Why do you choose the first 100 years? Models need time to reach new climate equilibrations in response to external forcings. I would use the last 100 years if possible at all.

**Response:** Thanks for the suggestion. This has been justified for reviewer 1 and 2 as well. We are using the first 100 years due to the reason that the simulations public on ESGF are already in equilibrium so there will be no significant differences for whether it is the first or the last 100 years. In many cases, only 100 years were available from ESGF. We now explain this more clearly in Section 2.2.

185: specify it is austral summer/winter. I also think this is where you can describe the regional climate systems in more detail. i.e., winter precipitation in the south is associated with the westerlies, summer precipitation in the north is associated with the monsoon.

**Response:** Thanks for the suggestion. We have expanded the description of the regional climate systems and specified austral summer/winter.

241: If "land areas warm more than surrounding oceans" during DJF and SON is the case, why DJF and SON show opposite signs in temperature change over Sahul? Are there other mechanisms that could cause this change in temperature?

**Response:** The paragraphs discussing seasonal temperature change have been rewritten to clarify the results. There were a number of points which required better explanation.

245-250: How do these analyses relate to your results in Figure 5? If there is enhanced cooling in SON and reduced cooling in MAM, why Fig 5 shows more cooling in MAM and less cooling in SON?

**Response:** This paragraph has been deleted as the insolation anomalies are not helpful in explaining the seasonal temperature anomalies.

311: What is this "SST gradient"?

**Response:** This sentence was deleted.

395-396: This statement does not make sense. Fig 5 shows DJF cooling and SON warming over northern Australia, why does it case wetting in both seasons? What is the "response to changes in seasonal heating" and "changes in atmospheric circulation" here?

**Response**: The discussion of drivers of change in rainfall has been rewritten to clarify. New figures showing changes in 850 hPa winds in Section 3.2 (new Figures 6 and 7) assist to show links with changes in offshore/onshore circulation and wind convergence.

414: p = 0.082 suggests that the correlation is not significant or "moderate" – it is insignificant. By the way, I wonder how do changes in precipitation and the northward displacement of easterly-westerly boundary correlate.

According to your findings, what is the mechanism for changes in winds?

**Response:** Thank you for this comment. We have now rewritten this section to indicate that the correlation is not statistically significant at the 95% confidence level. We also add a broader discussion of changes in westerlies and Southern Australian precipitation.

**Technical corrections**

268: You don't need a 3.2.1 subsection here

**Response:** This was removed as suggested.

323: Figure S4 is MMM seasonal anomalies for LGM - PI evapotranspiration, not precipitation.

**Response:** This was corrected.

397: to the => to the

**Response:** This was corrected.

403: should be 3.3.2.2

**Response:** This was corrected.

---

## Referee Report (RR1)

**General comments**

This manuscript by Du et al. has been much improved since first submitted, thanks to the diligent commitment of the authors to address the reviewers' comments. The paper reads easily and the parts which were rewritten have clarified the overall structure of the manuscript. I think that this modelling intercomparison study at the LGM is well suited for publication in Climate of the Past, providing an interesting focus on the Southern Hemisphere hydroclimate thanks to the case study on Australia.

I do have a few additionnal comments to hopefully guide further improvement. Some I have still classified as major.

**Major comments**

1. Knowledge gap: (a) the knowledge gap outlined in the abstract (L7 : « The climate changes… remain uncertain. ») is extremely laconic. I would argue that the abstract's objective is also to convince the reader to continue reading, and thus to explicitly present him or her with a problem worth solving. So I would recommend elaborating a bit on the knowledge gap in the abstract as well. [If the authors are limited by abstract length requirements, I think that the relationship between CMIP5 and 6 models and PMIP phase 3 and 4 models is comparatively much less important (and could be explained in Section 1.2 only).]
(b) The introduction sentence of the knowledge gap on L170-171" The LGM is commonly recognised as a time of global cooling and lower sea levels, best estimates placing this at ca. 21 ka. However, ..." is extremely confusing until the temporal discrepancy is pointed out later in L.174. The knowledge gap is also underdeveloped, to my opinion. Could the authors clarify and elaborate on the knowledge gap, possibly describing the different temporality of the SH regarding the start of the deglaciation, the bipolar seesaw mechanism, etc...?

2. Ending note (L20-22 and L1284-1289): I am not a fan of the 'further analysis is required' statements as it doesn't spell out clear directions of where the research should move forward to make progress on the still unresolved knowledge gaps of the paper. It is a bit of a shame to end the abstract and conclusion on an underwhelming note. Could you maybe provide clearer recommendations for modellers and for experimentalists, based on what we have learnt in this study? To better identify model biases – and better resolve mechanisms, what analysis are we lacking? Which sensitivity tests could be made? As for data, what do we need from data to better constrain models in the Australian regions?

3. L141-145 and Figure 1 / L191 : (a) Placement: The 'Australia case study' is brought to the reader's attention too soon, before it is even justified, and before starting describing the LGM overall climate again. In addition, Fig. 1 shows the different regions that are starting becoming relevant from Section 1.1 onwards. I would advise moving this figure to later in the text.
(b) Additional proxy information : while useful as it is, Fig. 1 could also be enriched with, e.g., the coring locations of all the proxies described in Section 1.1.
(c) Justifying the regionalisation : finally, I would like to point out that the connection between the separation into 3 regions and the atmospheric circulation mechanisms explaining the existence of this specific regionalisation is not expicitly made (neither in the legend of Fig 1, in L191, or around L175-177), until the much too late Section 2.3.
Hence, I would suggest reorganizing things (Fig. 1 / Section 1.1 / Section 2.3) so that the relevancy of this regionalisation becomes apparent to the reader in a logical manner.

4. The land-sea masks and their potential impacts. (a) It is a bit of the shame that Fig. 1 doesn't show the difference between PI and LGM land-sea masks with different contours.

(b) The authors have indicated that they will show the LGM land-sea masks for individual models in SI. I would suggest it may also be relevant to show those with additional contours in Fig. 3 and 9 notably, for I have been constantly wondering about the impact of different coastlines on the simulated variables and their potential disagreements. When the 'maritime' continent is becoming less maritime, leading to less evaporation, how does that affect the precipitation patterns over the whole region? My point is that different models, of different resolution, and using different ice-sheet reconstructions for the LGM, are not likely to have implemented the exact same coastlines e.g. around the Sahul shelf, leading to possible model disagreement in this region (i.e. stippling where the coastlines differ). I would like for the reader to have a chance to examine this potential effect (and if some shows up, for the authors to also discuss this).

**Specific comments**

- L12-13 « with a multimodel mean 2.9°C decrease in annual average surface air temperature over land at the LGM compared to the pre-industrial » is so packed with information that it is a bit difficult to read.

- L13-14 « while models show consistent patterns of regional cooling » confused me at first as it felt like a repetition of L11-12. Can't both of these informations be presented all at once ?

- L16-18 « […] vary greatly between modes […] shows little change […] are also uncertain, with wide model disagreement » : I was confused by the 'also' L18 since a sentence describing surface moisture balance changes was placed between the two sentences pointing out the model disagreement.

- L296: While the newly added summary paragraph works well, I would add here a transition sentence to the next section to justify the use of models to complete the picture formed with the proxy data, something along the lines of : "In this context, climate models could thus provide precious insights into the mechanisms responsible for this observed climate."

- Table 1: While the use of the first 100 years of model outputs has been justified in the reviews and in the text, I would suggest avoiding the terms "run length" and "length of simulation" (e.g. in Table 1 or in Section 2.2 title) to refer to the length of the model outputs available on the ESGF. The authors could use something like "output length after spin-up" or some other equivalent so as not to confuse the reader. I would even argue that actually, the third column of Table 1 is irrelevant, for (1) the authors are using only the first 100 years anyway, and (2) only the spin-up duration can give the reader an idea of how well equilibrated the LGM simulations are. So the authors could consider replacing this column with the spin-up duration numbers, or removing it altogether.

- Fig. 2: I admit to finding the he first occurence of the 70% stippling peculiar, for their is no stippling appearing in Fig. 2. The authors could either choose a higher standard (e.g. good agreement on the amplitude of the change?), or simply warn the reader with e.g. "A stippling indicating areas where less than 70% of ensemble members agree on the sign of the anomaly has been chosen, consistently with the following figures. As a result of the high agreement between models in terms of the sign of the temperature anomaly, no stippling is shown here."

- Fig. 5: Please consider further commenting on Fig. 5 in the text. What do we see in terms of model disagreement? Do we see an increased seasonality at the LGM wrt. PI? The authors could also consider quantifying the model spread or commenting on the obvious two outliers (one PMIP3 and one PMIP4) in the global mean temperature plot.

- L696: Please consider adding a transition sentence to connect the previous section with the one starting now.

- Fig. 6: I am wondering whether showing wind **changes** as vectors is the best data vizualisation choice. The reader may assume that arrows stand for the winds themselves.

- L793-795: Please also mention that some models do not show any significant change. Also, I would start with the change of westerly strength (L796) before mentioning the latitudinal shifts, as the latter is not the first metric that usually comes to mind.

- L829 "as noted in Section 3.2": It was not clearly spelled out in Section 3.2. I would suggest reformulating the transition, so that the writing can flow more in-between sections.

- L952: This mention to Table 3 may confuse the reader as to whether he should read Table 3 or Figure 12 first. I suggest it is not necessary here.

- L1063 "While a decomposition of thermodynamic and dynamic drying components of precipitation change is not included in this study, it is evident that the thermodynamic drying response is dominant": (a) Why not? It would be interesting to see such an analysis.
(b) On what basis can you say it is dominant (without doing the decomposition)? I may have missed the 'evident' fact which helps conclude this. Please justify, or nuance (for this is a strong statement, comparatively to e.g. L1059 "it is likely that...").

- L1230: (a) It feels like the winds subsection lacks a clear conclusion, along the lines of "This also reveals that much progress remains to be made with respect to...".
(b) Please also consider refering to this recent paper by Gray et al. (2023):
https://doi.org/10.1029/2023PA004666, to elaborate on your discussion.

- L1279-1280 "again suggesting that caution is required" is redundant with a previous statement, so it does not bring anything to the table. I suggest removing it.

**Technical comments**

- L139-140: Why not use 6.1 ± 0.4 °C, to provide uniformity with the previous figures?

- L305 "a slight increase" of what?

- In L315 and a couple of other times, the poleward/equatorward shifts are refered to as southward /northward. I think it is best to stick to poleward/equatorward even if the study is focusing only on the SH.

- L316 equatorward shifts (reversed word order)

- L318 "more recent" repetition

- L358 "the" -> their

- L1142 "has evaluated" -> evaluates

- L1158  "the models may not have resolved" -> the models do not resolve

---

## Referee Report (RR2)

Thanks to the authors' diligent and continuing efforts to address the reviewer's comments, I find that the manuscript is much improved since it was first submitted.

However, I still have recommendations, some related to potential clarification of new additions, and others pointing out the few but important sentences where the authors could make a more compelling case. In particular, I do not share the same view as the authors in their concluding remarks, as they are formulated now. I am detailing these recommendations below.

**General comments**

1. I find that some of the vocabulary related to climate models used in this paper is not always adequate or precise.

> - L10: '13 climate models that were included… are used to investigate...' → '13 climate model simulations...' or 'outputs'. Otherwise the formulation implies that the authors ran the simulations themselves.
>
> - L12: 'the model simulations are compared with existing proxy records...' → 'the simulated variables are compared…' or 'the variables of interest in simulations are compared'
>
> - L19: 'LGM precipitation anomalies are simulated differently between models' → 'models show   different…'. Otherwise it could imply that the precipitation processes are represented differently in the models.
>
> - L15: 'models do not have a robust response' → 'model simulations do not show a robust response'
>
> - L181: 'Many modelling studies have focused on the LGM as this period is one of the main 'entry card' experiments for PMIP'. The relevancy of this statement is questionable, as it reverses cause and consequence. There are scientific reasons why modellers are interested in simulating the LGM climate (which is why it has been defined as an entry card in PMIP).
>
> -L211: 'Evidence' is a slightly strong word for a simulated change. I would use a synonym (e.g. 'signs') to avoid confusion.

2. Although the authors have clarified the knowledge gap, the way it is formulated is not very compelling.

> - L8: 'remain uncertain': According to what: proxy studies?
>
> - L8-9: 'including the [list] of changes]': Is there a way to clarify the scientific problem you are tackling? This formulation using a list introduces a disconnection between your variable of interest. On the contrary, the emphasis could be on how these variables are connected (why you want to examine them together in the same paper), which tell us something about the processes.
>
> - L60: 'Questions about the climate of Australia include [list]'. Same as before, this is not a very compelling way to phrase a scientific question. Perhaps a more efficient way of telling the reader this would be to do it later in the paragraph, by explicitly relating this statement with L65-66 which mentions processes.

3. I remain unconvinced by the perspectives.

- L20: 'suggesting that caution is required when interpreting model output'. While true, this is also an obvious statement, so probably a wasted opportunity to teach something new to the reader. I find your new element of conclusion about different land-sea mask over Sahul to be much more insightful for modellers, and possibly worth a mention in the abstract.

- L22: 'is required to determine the drivers… and to identify the most plausible set of LGM simulations'. This statement puts on the same level a process-understanding objective ('determine the drivers') and a second part implying in its formulation that the end goal is to exclude some models based on their performance to get some kind of realistic ensemble. I would argue that the whole point of a multimodel comparison study is not to elect Mr. Best Model Out There (for all models are wrong, although some may give better results than others depending on the variables we are looking at) or exclude outliers, but really to use the model evaluation and the evidenced biases to learn something about the processes and determine how the model representation could be improved. **Same remark for L792-793**

- L780-781: I do not understand what the authors mean by this statement. Isn't quantitative model-data comparison a way of caracterizing model biases (and thus the 'uncertainty')?

- L789-790: Please justify this statement, as it is the first mention of a potential use for future climate. Alternatively, this could be justified in the introduction when outlining the aim of the present study.

4. The addition of the different LGM land-sea masks is indeed interesting to see. I have a few related recommendations.

- Figures 1, 4 and 9: I wonder if the use of an interpolated contour for the LGM land-sea mask makes the most sense. Possibly, a sharp delimitation (without interpolation) between the wet and dry grid cells would let the reader see more clearly the model mesh and resolution. However, it might deteriorate visibility.

- For consistency, Figures 1, 4 and 9 should also plot the pre-industrial land-sea masks in thin lines, and not just the high-resolution modern topography.

- L360 'the thin black lines indicate modern coastlines': Same, we cannot assume that the pre-industrial land-sea mask from individual model will follow well the high-resolution modern coastlines.

- L544-546: There is no mention of the sharp precipitation gradient that some models simulate at their coastlines (but not others). This could be worth mentioning.

- L755-756: As evidenced by the different land-sea mask, the model response may also be related to different boundary conditions (and not just a different model response to the change in boundary conditions).

**Specific comments**

L52-53: 'of lower temperatures… that cooled the climate'. This is more or less a repetition of the same element.

L58: 'While many studies'. Proxy studies or modelling studies? Please specify.

L58: 'globally and in the Northern Hemisphere'. Is this an 'and/or'? This sounds a bit contradictory.

L73: 'have begun to explore' → 'have explored'

L75: A link word and a recapitulation of the originality of the study would be welcomed to contrast this study with previous ones. Something along the lines of: 'Hence, the present study used the most recent PMIP3 and PMIP4 simulations to investigate climate changes over the Australian region specifically'.

L137: Mentioning the Gray et al., 2023 paper (and contrasting its conclusions to Kohfeld's) would be welcomed here.

L193-194 and L205-207: It is a bit confusing for the reader to switch abruptly (without link words) from the description of previous studies to the aims of the present study.

L199-204: Mentioning the changes in simulated SH westerly winds in the Gray et al., 2023 paper would also be welcomed here, as it is the most recent study.

Table 1 legend: The sentence related to length of simulation should be deleted (as the column was also deleted).

L252: Please delete 'minor'.

L257: 'PMIP3 ice-sheet configurations' → 'the PMIP3 ice-sheet configuration'

L262: 'with CMIP5 CNRM-CM5 having the smallest expander land area'. It is difficult to see why this statement is relevant here. Possibly, quantifying the range of the difference of surface area between PI and LGM (in km²), from the model with the smallest difference to the one with the largest would be more meaningful.

L296 'According to the PMIP protocols' and L297 'see Kageyama et al., 2017 for details of the spin-up protocol'. Why are you mentioning the recommendations (which are not always thoroughly followed) instead of the spin-up duration that was actually done for PMIP4 simulations (in Kageyama et al., 2021)?

L246: 'Otherwise' → 'Hence' or synonym

---

## Author Response (AR2)

**Response to Review 1:**

This study analyzed the simulated LGM climate relative to the pre-industrial climate in PMIP3 and PMIP4 models with a focus on Australia. The results show that although the models all simulate widespread cooling over this region, they are not consistent in simulating changes in precipitation large scale circulation patterns. Compared to the last version, this manuscript is much improved, with more insightful analysis and discussion on the mechanisms driving hydroclimate change in Australia during the LGM, as well as more robust statistical analyses. However, for the issues I listed below, I suggest that the manuscript undergo another round of revision.

The authors tried to perform a model and data comparison. However, it remains unclear which models agree better with the proxy data, which leads to a rather open conclusion at the end. In my opinion, perhaps the model-data comparison could be discussed in more detail. For example, in the paragraph of line 520, the authors suggest that the proxy-inferred moisture availability is not consistent with changes in P-E from the PMIP models. From Figures 11 and 12 we see that the models do not agree on the sign of P-E over much of Australia. Therefore, I expect some models might agree with the proxy data better than the others for these particular regions.

**Response:** Thank you for the comments, we have rewritten the discussion in Section 4.2 to include a more detailed model-data comparison at a regional scale.

**Other comments:**

20: This statement is a little odd. We can use the model-proxy data comparison approach to determine which simulations perform the best. Then by analyzing this set of simulations, we can determine the driver of circulation changes during the LGM.

**Response:** This last two sentences in the Abstract have been edited to reflect the possible role of model-proxy comparison in selecting the best performing model.

342: There are 12 models, but only 5 models are described for changes in westerlies. What about the other 7?

**Response:** This paragraph has been expanded to include more discussion of all the models used in the study.

364: Are there dynamic components that may cause the "drying" of the Australian mainland? e.g., shifting of the westerlies away from the region?

**Response:** This paragraph has now been expanded to include discussions relating to dynamic changes that may be associated with the precipitation patterns simulated over each region.

485: Here you show that the westerly wind and precipitation are weakly correlated, but

what about the meridional shifts in westerlies? You mentioned that some models show a southward shift while others show a northward shift. Do these shifts cause consistent changes in precipitation in the respective models?

**Response:** Thank you for the suggestion. We have added the plot of latitudinal shift of maximum westerly winds versus precipitation in JJA in models as Supplementary Figure S9b and added a comment in the text to note the weak correlation between westerly wind position and precipitation. We also note that there is no significant correlation at the 95% confidence level, new methods could be used to quantify the meridional shifts in SH westerlies at the LGM (e.g. as proposed by Gray et al., 2023).

Reference:
Gray, W. R., de Lavergne, C., Jnglin Wills, R. C., Menviel, L., Spence, P., Holzer, M., Kageyama, M., and Michel, E.: Poleward shift in the Southern Hemisphere westerly winds synchronous with the deglacial rise in $CO_2$, Paleoceanogr. Paleoclimatol., 38, e2023PA004666, https://doi.org/10.1029/2023PA004666, 2023.

**Response to Review 2:**

**General comments**

This manuscript by Du et al. has been much improved since first submitted, thanks to the diligent commitment of the authors to address the reviewers' comments. The paper reads easily and the parts which were rewritten have clarified the overall structure of the manuscript. I think that this modelling intercomparison study at the LGM is well suited for publication in Climate of the Past, providing an interesting focus on the Southern Hemisphere hydroclimate thanks to the case study on Australia.

I do have a few additional comments to hopefully guide further improvement. Some I have still classified as major.

**Major comments**

1. Knowledge gap:

(a) the knowledge gap outlined in the abstract (L7 : « The climate changes... remain uncertain. ») is extremely laconic. I would argue that the abstract's objective is also to convince the reader to continue reading, and thus to explicitly present him or her with a problem worth solving. So I would recommend elaborating a bit on the knowledge gap in the abstract as well. [If the authors are limited by abstract length requirements, I think that the relationship between CMIP5 and 6 models and PMIP phase 3 and 4 models is comparatively much less important (and could be explained in Section 1.2 only).]

**Response:** Thank you for the suggestion. We have expanded the explanation of the knowledge gaps and edited the Abstract as suggested.

(b) The introduction sentence of the knowledge gap on L170-171" The LGM is commonly recognised as a time of global cooling and lower sea levels, best estimates placing this at ca. 21 ka. However, ..." is extremely confusing until the temporal discrepancy is pointed out later in L.174. The knowledge gap is also underdeveloped, to my opinion. Could the authors clarify and elaborate on the knowledge gap, possibly describing the different temporality of the SH regarding the start of the deglaciation, the bipolar seesaw mechanism, etc...?

**Response:** The knowledge gap has been expanded now, and the discussion of the timing of the LGM in Australia has been removed from this section to avoid confusion.

2. Ending note (L20-22 and L1284-1289): I am not a fan of the 'further analysis is required' statements as it doesn't spell out clear directions of where the research should move forward to make progress on the still unresolved knowledge gaps of the paper. It is a bit of a shame to end the abstract and conclusion on an underwhelming note. Could you maybe provide clearer recommendations for modellers and for experimentalists, based on what we have learnt in this study? To better identify model biases – and better resolve mechanisms, what analysis are we lacking? Which sensitivity tests could be made? As for data, what do we need from data to better constrain models in the Australian regions?

**Response:** The abstract has been rewritten to more clearly identify the nature of future work required. The conclusions have also been expanded to address some of the comments of the reviewer. However, we feel that as this is a preliminary study, we are not able to fully map out future research directions.

3. L141-145 and Figure 1 / L191 : (a) Placement: The 'Australia case study' is brought to the reader's attention too soon, before it is even justified, and before starting describing the LGM overall climate again. In addition, Fig. 1 shows the different regions that are starting becoming relevant from Section 1.1 onwards. I would advise moving this figure to later in the text.

**Response:** The figure has been moved to a later position.

(b) Additional proxy information : while useful as it is, Fig. 1 could also be enriched with, e.g., the coring locations of all the proxies described in Section 1.1.

**Response:** We have added the main locations of the proxies into Figure 1, showing by dots with numbers. The other proxy records without specific indication of locations are described in the text in Section 1.1. As this paper does not focus on proxy records, more detailed discussion of proxy records of LGM climate in the SH can be found in Petherick et al. (2022), which we now cite.

Reference:

Petherick, L. M., Knight, J., Shulmeister, J., Bostock, H., Lorrey, A., Fitchett, J., Eaves, S., Vandergoes, M. J., Barrows, T. T., Barrell, D. J. A., Eze, P. N., Hesse, P., Jara, I. A., Mills, S., Newnham, R., Pedro, J., Ryan, M., Saunders, K. M., White, D., Rojas, M., and Turney, C.: An extended last glacial maximum in the Southern Hemisphere: A contribution to the SHeMax project, Earth Sci Rev., 231, 104090, https://doi.org/10.1016/j.earscirev.2022.104090, 2022.

(c) Justifying the regionalisation : finally, I would like to point out that the connection between the separation into 3 regions and the atmospheric circulation mechanisms explaining the existence of this specific regionalisation is not expicitly made (neither in the legend of Fig 1, in L191, or around L175-177), until the much too late Section 2.3. Hence, I would suggest reorganizing things (Fig. 1 / Section 1.1 / Section 2.3) so that the relevancy of this regionalisation becomes apparent to the reader in a logical manner.

**Response:** This issue arises due to our responses to previous reviewer recommendations to combine figures and rearrange material. To resolve the issue, we have moved Figure 1 to the Data and Methods section 2.3 where it more clearly belongs. In Section 1.1 we now refer more generally to the proxy records within three climate zones, consistent with previous work (e.g. Reeves et al. 2013a; Petherick et al. 2013; Fitzsimmons et al. 2013). We think this is a more logical arrangement of the material.

4. The land-sea masks and their potential impacts. (a) It is a bit of the shame that Fig. 1 doesn't show the difference between PI and LGM land-sea masks with different contours.

**Response:** The LGM land mask contour has been added to Figure 1.

(b) The authors have indicated that they will show the LGM land-sea masks for individual models in SI. I would suggest it may also be relevant to show those with additional contours in Fig. 3 and 9 notably, for I have been constantly wondering about the impact of different coastlines on the simulated variables and their potential disagreements. When the 'maritime' continent is becoming less maritime, leading to less evaporation, how does that affect the precipitation patterns over the whole region? My point is that different models, of different resolution, and using different ice- sheet reconstructions for the LGM, are not likely to have implemented the exact same coastlines e.g. around the Sahul shelf, leading to possible model disagreement in this region (i.e. stippling where the coastlines differ). I would like for the reader to have a chance to examine this potential effect (and if some shows up, for the authors to also discuss this).

**Response:** The new figs 3 and 9 has been made with the black thick lines indicating the LGM coastlines prescribed in different modern, and the thin black line for modern coastlines. Corresponding discussion is also added in Sections 3.1 and 3.3.

**Specific comments**

- L12-13 « with a multimodel mean 2.9°C decrease in annual average surface air temperature over land at the LGM compared to the pre-industrial » is so packed with information that it is a bit difficult to read.

**Response:** This was rewritten.

- L13-14 « while models show consistent patterns of regional cooling » confused me at first as it felt like a repetition of L11-12. Can't both of these informations be presented all at once ?

**Response:** This was rewritten.

- L16-18 « [...] vary greatly between modes [...] shows little change [...] are also uncertain, with wide model disagreement » : I was confused by the 'also' L18 since a sentence describing surface moisture balance changes was placed between the two sentences pointing out the model disagreement.

**Response:** "Also" was deleted.

- L296: While the newly added summary paragraph works well, I would add here a transition sentence to the next section to justify the use of models to complete the picture formed with the proxy data, something along the lines of : "In this context, climate models could thus provide precious insights into the mechanisms responsible for this observed climate."

**Response:** A transition sentence was added.

- Table 1: While the use of the first 100 years of model outputs has been justified in the reviews and in the text, I would suggest avoiding the terms "run length" and "length of simulation" (e.g. in Table 1 or in Section 2.2 title) to refer to the length of the model outputs available on the ESGF. The authors could use something like "output length after spin-up" or some other equivalent so as not to confuse the reader. I would even argue that actually, the third column of Table 1 is irrelevant, for (1) the authors are using only the first 100 years anyway, and (2) only the spin-up duration can give the reader an idea of how well equilibrated the LGM simulations are. So the authors could consider replacing this column with the spin-up duration numbers, or removing it altogether.

**Response:** The title for Section 2.2 has been changed, and the third column in Table 1 has been deleted.

- Fig. 2: I admit to finding the he first occurence of the 70% stippling peculiar, for their is no stippling appearing in Fig. 2. The authors could either choose a higher standard (e.g. good agreement on the amplitude of the change?), or simply warn the reader with e.g. "A stippling indicating areas where less than 70% of ensemble members agree on the sign of the anomaly has been chosen, consistently with the following figures. As a result of the high agreement between models in terms of the sign of the temperature anomaly, no stippling is shown here."

**Response:** A sentence was added to clarify the confusion with no stippling in Fig 2 as suggested.

- Fig. 5: Please consider further commenting on Fig. 5 in the text. What do we see in terms of model disagreement? Do we see an increased seasonality at the LGM wrt. PI? The authors could also consider quantifying the model spread or commenting on the obvious two outliers (one PMIP3 and one PMIP4) in the global mean temperature plot.

**Response:** More discussion regarding Fig 5 was added in Section 3.1.

- L696: Please consider adding a transition sentence to connect the previous section with the one starting now.

**Response:** A transition sentence was added.

- Fig. 6: I am wondering whether showing wind **changes** as vectors is the best data vizualisation choice. The reader may assume that arrows stand for the winds themselves.

**Response:** For consistency with all other plots, we show the changes in winds rather than the actual winds. This is also consistent with other papers on the LGM simulations, e.g. Yan et al. (2018) plot wind vector changes.

- L793-795: Please also mention that some models do not show any significant change. Also, I would start with the change of westerly strength (L796) before mentioning the latitudinal shifts, as the latter is not the first metric that usually comes to mind.

**Response:** The entire paragraph was reframed.

- L829 "as noted in Section 3.2": It was not clearly spelled out in Section 3.2. I would suggest reformulating the transition, so that the writing can flow more in-between sections.

**Response:** This was a typo and should have referred back to Section 2.3 where the different climatic regions were first discussed. This has been reworded.

- L952: This mention to Table 3 may confuse the reader as to whether he should read Table 3 or Figure 12 first. I suggest it is not necessary here.

**Response:** This sentence referring to Table 3 has been removed.

- L1063 "While a decomposition of thermodynamic and dynamic drying components of precipitation change is not included in this study, it is evident that the thermodynamic drying response is dominant": (a) Why not? It would be interesting to see such an analysis. (b) On what basis can you say it is dominant (without doing the decomposition)? I may have missed the 'evident' fact which helps conclude this. Please justify, or nuance (for this is a strong statement, comparatively to e.g. L1059 "it is likely that...").

**Response:** The thermodynamic response would be drying due to cooler temperatures (consistent with Clausius-Clapeyron). The sentence was reframed to avoid confusion.

- L1230: (a) It feels like the winds subsection lacks a clear conclusion, along the lines of "This also reveals that much progress remains to be made with respect to...".
(b) Please also consider refering to this recent paper by Gray et al. (2023): https://doi.org/10.1029/2023PA004666, to elaborate on your discussion.

**Response:** The paper and the conclusion sentence were added.

- L1279-1280 "again suggesting that caution is required" is redundant with a previous statement, so it does not bring anything to the table. I suggest removing it.

**Response:** This has been removed.

**Technical comments**

- L139-140: Why not use 6.1 ± 0.4 °C, to provide uniformity with the previous figures?

**Response:** This has been modified.

- L305 "a slight increase" of what?

**Response:** This sentence has been modified.

- In L315 and a couple of other times, the poleward/equatorward shifts are refered to as southward /northward. I think it is best to stick to poleward/equatorward even if the study is focusing only on the SH.

**Response:** This was corrected.

- L316 equatorward shifts (reversed word order)

**Response:** This has been modified.

- L318 "more recent" repetition

**Response:** This has been modified.

- L358 "the" -> their

**Response:** This has been modified.

- L1142 "has evaluated" -> evaluates

**Response:** This has been modified.

- L1158 "the models may not have resolved" -> the models do not resolve

**Response:** This has been modified.

**Response to Review 3:**

The authors compared models simulations with existing proxy to investigate the climate changes at the LGM over the Australian region, in terms of temperature, precipitation, moisture balance and wind. Compared to the previous version, the current version of the manuscript by Du et al. has been improved and well written. Here are a few basic comments and reviews, mainly on the SH westerlies, for the authors' consideration.

**Comments:**

1. In Figure 8, the strength of westerly is calculated over all longitudes. But the regional westerly over Australia would be much better since the regional westerly is closely related to the precipitation over the southwest of Australia (as shown in figure S9).

   **Response:** We have replotted Figure 8 to just calculate the strength of westerly winds over Australian longitudes defined in the study (110°E-160°E).

2. In section 4.3, there is no proxy or evidence to illustrate the behavior of the westerlies in LGM. Is there any proxy to indicate the strength or the position of the mid-latitude westerlies? To constraint the characteristics of westerlies in LGM, it would be useful to collect proxy or geological evidence in future, especially the direct evidence, such as the grain sizes of eolian sediments.

   **Response:** We have added future suggestions in Section 4.3 for using the new method of calculating the SST front latitude in climate models to quantify shifts in the westerlies at the LGM, proposed by Gray et al. (2023).

3. For the southern westerlies change, the simulated sea ice is important to explain the westerlies over the Southern Hemisphere (Chavaillaz et al., 2013). Thus, the authors might consider to analyze the simulated sea ice to illustrate the diversity of the westerlies in LGM among CMIP models in future.

   Referrences:
   Chavaillaz, Y., Codron, F., and Kageyama, M.: Southern westerlies in LGM and future (RCP4.5) climates, 2013, Clim. Past, 9, 517–524,

   **Response:** Thanks for the suggestion. We have added the discussion of this reference in Section 4.3 as suggestions for future research.

**Minors:**

1. In Figure 5c and 5d, the value of MMM of PMIP4 models is larger than each ensemble. Please check the calculation.

   **Response:** Thanks for the reminder. We have checked the calculation, and they are correct.

2. Figures 6, 11 and 12, the texts for the title of each panel are too much and could be simplified. For example, 'LGM-PI' might be excluded.

**Response:** Thanks for the suggestion. We kept it there aiming to clarify that the figures are showing anomaly values, rather than LGM patterns.

3. Line 170, there are two 'only' here.

**Response:** This was corrected.

---

## Author Response (AR3)

Thanks to the authors' diligent and continuing efforts to address the reviewer's comments, I find that the manuscript is much improved since it was first submitted.

However, I still have recommendations, some related to potential clarification of new additions, and others pointing out the few but important sentences where the authors could make a more compelling case. In particular, I do not share the same view as the authors in their concluding remarks, as they are formulated now. I am detailing these recommendations below.

Thanks to the reviewer for their comments. We have revised the manuscript in response to the review except in a few cases where we disagree with the suggested changes.

**General comments**

1. I find that some of the vocabulary related to climate models used in this paper is not always adequate or precise.

- L10: '13 climate models that were included... are used to investigate...' → '13 climate model simulations...' or 'outputs'. Otherwise the formulation implies that the authors ran the simulations themselves.

**Response:** "simulations" was added after "climate models".

- L12: 'the model simulations are compared with existing proxy records...' → 'the simulated variables are compared...' or 'the variables of interest in simulations are compared'

**Response:** We do not think it is necessary to change the wording here. It is obvious that selected model variables are compared with proxy records and other model studies.

- L19: 'LGM precipitation anomalies are simulated differently between models' → 'models show different...'. Otherwise it could imply that the precipitation processes are represented differently in the models.

**Response:** The wording was changed to "Models simulate a range of LGM precipitation anomalies over the region".

- L15: 'models do not have a robust response' → 'model simulations do not show a robust response'

**Response:** The wording was changed as suggested.

- L181: 'Many modelling studies have focused on the LGM as this period is one of the main 'entry card' experiments for PMIP'. The relevancy of this statement is questionable, as it reverses cause and consequence. There are scientific reasons why modellers are interested in simulating the LGM climate (which is why it has been defined as an entry card in PMIP).

**Response:** This part of the sentence was deleted.

-L211: 'Evidence' is a slightly strong word for a simulated change. I would use a synonym (e.g. 'signs') to avoid confusion.

**Response:** (L147) "Evidence for change" was replaced with "Simulated changes".

2. Although the authors have clarified the knowledge gap, the way it is formulated is not very compelling.

- L8: 'remain uncertain': According to what: proxy studies?

**Response:** The uncertainty reflects disagreement in proxy studies as well as previous modelling studies. This sentence is an introductory one, with the details provided in the Introduction section of the paper. We think the existing wording should be sufficiently clear to the reader.

- L8-9: 'including the [list] of changes]': Is there a way to clarify the scientific problem you are tackling? This formulation using a list introduces a disconnection between your variable of interest. On the contrary, the emphasis could be on how these variables are connected (why you want to examine them together in the same paper), which tell us something about the processes.

**Response:** The aim of this paper is to provide a preliminary overview of the climate of Australia as simulated in PMIP LGM simulations. The study does not aim to address one specific variable or aspect of the climate circulation, but rather summarise the extent of model to model agreement and the extent of agreement between models and proxy records. This will then hopefully stimulate further studies into particular aspects of the Australian region at the LGM (e.g. monsoon, ENSO, frontal systems, etc.), and will be of relevance for interpretation of proxy records. Focusing on several different variables and modes of variability is not uncommon for such overview studies, e.g. see Grose et al. (2020) study of Australian climate in CMIP5 and CMIP6 models.

Reference: Grose, M. R., Narsey, S., Delage, F. P., Dowdy, A. J., Bador, M., Boschat, G., ... & Power, S. (2020). Insights from CMIP6 for Australia's future climate. Earth's Future, 8(5), e2019EF001469.

- L60: 'Questions about the climate of Australia include [list]'. Same as before, this is not a very compelling way to phrase a scientific question. Perhaps a more efficient way of telling the reader this would be to do it later in the paragraph, by explicitly relating this statement with L65-66 which mentions processes.

**Response:** Please note that this sentence was added in response to a previous review. We think the current version is adequate to introduce the themes of the paper. Any further edits here appear to be personal stylistic preferences of the reviewer, rather than relating to the scientific content or clarity of the paper. We feel that further editing of this paragraph is scientifically unnecessary and may not improve the manuscript, but we will follow the Editor's guidance.

3. I remain unconvinced by the perspectives.

**General response:** While we appreciate the reviewer's suggestions, we feel that it is appropriate for us as the authors to determine our own conclusions and perspectives. We are happy to follow the Editor's guidance and make revisions if there are scientific errors or the text is unclear.

- L20: 'suggesting that caution is required when interpreting model output'. While true, this is also an obvious statement, so probably a wasted opportunity to teach something new to the reader. I find your new element of conclusion about different land-sea mask over Sahul to be much more insightful for modellers, and possibly worth a mention in the abstract.

**Response:** We wished to provide a warning to those using output from climate models to compare with proxy records, particularly when relying on a single model. We feel that this general point is still relevant so we would like to retain it here. Regarding the land-sea mask, this is a relatively technical conclusion so we feel it would be better addressed in the Conclusions (following explanation of the LGM land masks in the models) rather than presented out of context in the Abstract.

- L22: 'is required to determine the drivers... and to identify the most plausible set of LGM simulations'. This statement puts on the same level a process-understanding objective ('determine the drivers') and a second part implying in its formulation that the end goal is to exclude some models based on their performance to get some kind of realistic ensemble. I would argue that the whole point of a multimodel comparison study is not to elect Mr. Best Model Out There (for all models are wrong, although some may give better results than others depending on the variables we are looking at) or exclude outliers, but really to use the model evaluation and the evidenced biases to learn something about the processes and determine how the model representation could be improved. **Same remark for L792-793**

**Response:** There are many points of view on the goals of palaeoclimate modelling studies, including to evaluate and improve models as well as to better understand the climate of the past (as discussed in Kageyama et al. 2017 and numerous other papers). Given this, we attempted to address both sets of goals with our concluding statement. We have now reworded the sentence to the following to remove reference to the most plausible set of model simulations:

*"Further analysis based on model evaluation and quantitative model-proxy comparison is required to better understand the drivers of LGM climate and atmospheric circulation changes in this region."*

At line 792-793: We think the discussion in Section 4.4 is reasonable, as it includes both perspectives based on better understanding past climate via modelling studies as well as possibly selecting models with a better simulation of regional climate. This is consistent with the range of perspectives in the climate and palaeoclimate modelling communities and we do not think further revision of this section of the manuscript is needed.

- L780-781: I do not understand what the authors mean by this statement. Isn't quantitative model-data comparison a way of caracterizing model biases (and thus the 'uncertainty')?

**Response:** We intended to state that given large uncertainty (including over the sign of change) in both models and proxies, it is not currently possible to make robust conclusions about whether it was wetter or drier over northern Australia at the LGM based on the proxy and model evidence.

This conclusion follows from the discussions in the paper - there is uncertainty in interpretation of hydroclimate proxy records due to offsetting changes in both precipitation and evaporation as well as $CO_2$ effects in vegetation proxies, so it is difficult to conclude that it was clearly wetter or drier in many regions. In addition, models simulate inconsistent precipitation changes, with some models simulating wetter and others simulating drier conditions over northern Australia.

We have attempted to rewrite this sentence to more clearly summarise our point.

*"Given the large uncertainty (including over the sign of change) in both models and proxies, it is not possible to make robust conclusions about whether it was wetter or drier at a regional scale over Northern Australia at the LGM based on the available evidence. "*

- L789-790: Please justify this statement, as it is the first mention of a potential use for future climate. Alternatively, this could be justified in the introduction when outlining the aim of the present study.

**Response:** A brief mention of the relevance of understanding past changes in the region for constraining uncertainty in future projections is now added in the Introduction with reference to two relevant papers (Grose et al., 2020 and Narsey et al., 2020). A more detailed discussion of the relevance of past modelling for future projections in this region is beyond the scope of the current study but will be addressed in future work.

4. The addition of the different LGM land-sea masks is indeed interesting to see. I have a few related recommendations.

- Figures 1, 4 and 9: I wonder if the use of an interpolated contour for the LGM land-sea mask makes the most sense. Possibly, a sharp delimitation (without interpolation) between the wet and dry grid cells would let the reader see more clearly the model mesh and resolution. However, it might deteriorate visibility.

**Response:** Thanks for the suggestion. However we have decided not to plot the land-sea mask in this way as it would be difficult to see the climate variables of interest in the same plots.

- For consistency, Figures 1, 4 and 9 should also plot the pre-industrial land-sea masks in thin lines, and not just the high-resolution modern topography.

**Response:** We do not think it is necessary to plot the pre-industrial land-sea masks as well as the LGM land masks. Numerous other papers describing PMIP LGM simulations use only modern coastlines or add the LGM coastlines but without including equivalent pre-industrial coastlines (e.g. Yan et al., 2018; Brown et al., 2020; Kageyama et al., 2021; Wang et al., 2023). We will follow the Editor's guidance on this matter.

- L360 'the thin black lines indicate modern coastlines': Same, we cannot assume that the pre-industrial land-sea mask from individual model will follow well the high-resolution modern coastlines.

**Response:** We do agree that the piControl land-sea mask configurations may differ across models and not exactly follow the high-resolution modern coastlines. However, we do not think this is a major consideration compared with the dramatic changes in LGM coastlines. We do not think it is necessary to include the pre-industrial land masks on the figures, as discussed in the response to the previous point.

- L544-546: There is no mention of the sharp precipitation gradient that some models simulate at their coastlines (but not others). This could be worth mentioning.

**Response:** One sentence has been added in Section 3.3 to mention this finding.

- L755-756: As evidenced by the different land-sea mask, the model response may also be related to different boundary conditions (and not just a different model response to the change in boundary conditions).

**Response:** Thanks for that. One sentence has been added in Section 4.4 to address it.

**Specific comments**

L52-53: 'of lower temperatures... that cooled the climate'. This is more or less a repetition of the same element.

**Response:** "Cooled the climate" was deleted.

L58: 'While many studies'. Proxy studies or modelling studies? Please specify.

**Response**: "modelling" added before "studies".

L58: 'globally and in the Northern Hemisphere'. Is this an 'and/or'? This sounds a bit contradictory.

**Response:** We are highlighting the lack of studies which focus on regional changes in the Southern Hemisphere, e.g. over Australia or other SH continents. This is not contradictory to other studies focusing on global changes e.g. Kageyama et al. 2021 and lots of NH regional studies.

L73: 'have begun to explore' → 'have explored'

**Response:** Changed as suggested.

L75: A link word and a recapitulation of the originality of the study would be welcomed to contrast this study with previous ones. Something along the lines of: 'Hence, the present study used the most recent PMIP3 and PMIP4 simulations to investigate climate changes over the Australian region specifically'.

**Response:** Edited as suggested.

L137: Mentioning the Gray et al., 2023 paper (and contrasting its conclusions to Kohfeld's) would be welcomed here.

**Response:** This has been added as suggested.

L193-194 and L205-207: It is a bit confusing for the reader to switch abruptly (without link words) from the description of previous studies to the aims of the present study.

**Response:** The sentences were refined to improve consistency.

L199-204: Mentioning the changes in simulated SH westerly winds in the Gray et al., 2023 paper would also be welcomed here, as it is the most recent study.

**Response:** This has been added as suggested in Section 1.2.

Table 1 legend: The sentence related to length of simulation should be deleted (as the column was also deleted).

**Response:** Deleted.

L252: Please delete 'minor'.

**Response:** Deleted.

L257: 'PMIP3 ice-sheet configurations' → 'the PMIP3 ice-sheet configuration'

**Response:** Changed.

L262: 'with CMIP5 CNRM-CM5 having the smallest expander land area'. It is difficult to see why this statement is relevant here. Possibly, quantifying the range of the difference of surface area between PI and LGM (in km2), from the model with the smallest difference to the one with the largest would be more meaningful.

**Response:** The statement was deleted.

L296 'According to the PMIP protocols' and L297 'see Kageyama et al., 2017 for details of the spin-up protocol'. Why are you mentioning the recommendations (which are not always thoroughly followed) instead of the spin-up duration that was actually done for PMIP4 simulations (in Kageyama et al., 2021)?

**Response:** It is relevant to mention the PMIP4 protocol here. We now add a reference to Kageyama et al. (2021) for the actual spin-up durations as suggested.

L246: 'Otherwise' → 'Hence' or synonym

**Response:** Changed to "Overall" as this more clearly indicates the lack of an overall correlation between global and regional temperature anomalies despite some matches.